# EMBODIED AGENTS MEET PERSONALIZATION: INVESTIGATING CHALLENGES AND SOLUTIONS THROUGH THE LENS OF MEMORY UTILIZATION

**Taeyoon Kwon**[*1]    **Dongwook Choi**[*1]    **Hyojun Kim**[1]    **Sunghwan Kim**[1]
**Seungjun Moon**[1]    **Beong-woo Kwak**[1]    **Kuan-Hao Huang**[2]    **Jinyoung Yeo**[1]
[1]Department of Artificial Intelligence, Yonsei University
[2]Department of Computer Science and Engineering, Texas A&M University

## ABSTRACT

LLM-powered embodied agents have shown success on conventional object-rearrangement tasks, but providing personalized assistance that leverages user-specific knowledge from past interactions presents new challenges. We investigate these challenges through the lens of agents' memory utilization along two critical dimensions: object semantics (identifying objects based on personal meaning) and user patterns (recalling sequences from behavioral routines). To assess these capabilities, we construct MEMENTO, an end-to-end two-stage evaluation framework comprising single-memory and joint-memory tasks. Our experiments reveal that current agents can recall simple object semantics but struggle to apply sequential user patterns to planning. Through in-depth analysis, we identify two critical bottlenecks: information overload and coordination failures when handling multiple memories. Based on these findings, we explore memory architectural approaches to address these challenges. Given our observation that episodic memory provides both personalized knowledge and in-context learning benefits, we design a hierarchical knowledge graph-based user-profile memory module that separately manages personalized knowledge, achieving substantial improvements on both single and joint-memory tasks. Our code and data is available at `https://github.com/Connoriginal/MEMENTO`.

## 1 INTRODUCTION

Embodied agents empowered by large language models (LLMs) have recently demonstrated remarkable success in executing object rearrangement tasks in household environments (Huang et al., 2022b; Wu et al., 2023; Song et al., 2023; Chang et al., 2024; Li et al., 2025b). As the primary objective of embodied agents is to provide assistance to users while interacting with the physical world, leveraging LLMs' natural language understanding and planning capabilities leads embodied agents to effectively interpret user instructions to successfully accomplish the task.

But do such tasks truly reflect the challenges in providing practical assistance to the users? As illustrated in Figure 1, conventional embodied tasks predominantly focus on single-turn interactions with static and simplified instructions that the agents could simply follow without implicit reasoning to comprehend user intentions (Ahn et al., 2022; Huang et al., 2022a; Song et al., 2023; Wu et al., 2024). However, for personalized embodied agents, it is important to understand personalized knowledge that users assign unique semantics to the physical world (*e.g.*, favorite cup, breakfast routine) to interpret dynamic instructions. To provide personalized assistance, agents must understand user instructions and conduct task planning by leveraging personalized knowledge retained from previous interactions—especially episodic memory, which enables the recall of specific events grounded in time and space (Huet et al., 2025). Therefore, understanding embodied agents' memory utilization capabilities is crucial for providing practical assistance to users.

In this work, we investigate the challenges faced by LLM-powered embodied agents in personalized assistance tasks through the lens of memory utilization. We analyze agents' memory utilization

---

*Equal contribution

Figure 1: Comparison between conventional embodied tasks and personalized assistance tasks. Previous works focus on following simple instructions, while personalized assistance agents must know user-specific knowledge, which require grounding in past interactions.

capabilities along two critical dimensions based on personalized knowledge: (1) the ability to identify target objects based on personal meaning (*object semantics*), and (2) the ability to recall sequences of object-location pairs from consistent user behavioral patterns, such as daily routines (*user patterns*). To evaluate task performance and quantify how memory utilization affects performance on personalized assistance tasks, we construct MEMENTO, an end-to-end two-stage personalized embodied agent evaluation framework comprising both single-memory and joint-memory tasks. Using MEMENTO, we address the following three research questions:

**RQ1: Can current LLM-powered embodied agents effectively utilize memory to perform personalized assistance tasks?** (Section 4) Initially, we evaluate various LLM-powered embodied agents using MEMENTO to assess their memory utilization capabilities. Our findings reveal a key limitation: while agents can effectively recall simple object-related memory for object semantics, they struggle to apply sequential information such as user patterns to their planning process. Qualitative analysis shows that conflicts between parametric commonsense knowledge and non-parametric personalized knowledge can cause agents to neglect memory utilization, indicating that current LLM-powered embodied agents cannot reliably integrate personalized knowledge for task execution.

**RQ2: What are the key factors that negatively affect embodied agents' memory utilization capabilities?** (Section 5) To understand the underlying factors that limit memory utilization performance, we conduct an in-depth analysis of agents' behavior under varying memory conditions. Our investigation reveals two critical bottlenecks: (1) *information overload*: increasing the number of retrieved memories (top-$k$) introduces noise that degrades performance, and (2) *coordination failures*: agents cannot effectively coordinate multiple memories, failing even on simple joint-memory tasks combining two object semantic memories. These findings underscore the essential need for improved memory architectures that support agents in personalized assistance tasks.

**RQ3: How can we design memory architectures to better support personalized assistance tasks?** (Section 6) Based on our findings, we explore architectural approaches to improve agents' memory utilization capabilities. Initially, we investigate memory summarization to remove unnecessary information but find marginal improvements and even degradation in smaller LLMs. This result reveals that episodic memory provides both personalized knowledge and in-context learning benefits, making it an essential memory module for embodied agents. Therefore, we develop a hierarchical knowledge graph-based *user-profile memory* module that manages personalized information independently. Adapting this memory module shows substantial improvements on both single and joint-memory tasks demonstrating the potential for future research in developing personalized embodied agents.

## 2 RELATED WORK

**LLM-powered embodied agents.** LLMs have significantly advanced embodied agents' reasoning and planning capabilities in recent years. Researchers have explored LLMs for interpreting user goals (Ahn et al., 2022), high-level task planning (Huang et al., 2022a), and integrating LLMs into comprehensive embodied agent frameworks (Huang et al., 2022b; Li et al., 2022; Mu et al., 2023; Song et al., 2023; Huang et al., 2023b). Other research directions have focused on generating

executable code for embodied tasks directly from language instructions (Liang et al., 2022; Wang et al., 2023a; Singh et al., 2023), while various benchmarks have been developed to evaluate embodied reasoning abilities (Li et al., 2023; Choi et al., 2024; Chang et al., 2024; Li et al., 2025b). Collectively, these studies highlight the promise of LLM-powered agents in bridging language understanding and physical interaction.

**Memory systems for embodied agents.** Previous studies on memory systems for embodied agents have primarily focused on semantic memory (*e.g.*, scene graph, semantic map), which store and provide state information about the current environment (Rana et al., 2023; Kim et al., 2023; Gu et al., 2024; Xie et al., 2024; Wang et al., 2024c), or on procedural memory (*e.g.*, skill library) that stores action primitives, focusing on how to perform tasks to enhance the efficiency in generating the low-level action code (Wang et al., 2023a; Sarch et al., 2023; Zhang et al., 2023). Another important category is *episodic memory*, which captures specific past interactions and experiences with users. However, prior uses of episodic memory have mostly treated it as passive task buffers (Ahn et al., 2022; Singh et al., 2023) or histories for in-context (Huang et al., 2022b; Liu et al., 2023; Song et al., 2023; Chen et al., 2024), without explicitly evaluating its role in personalized task grounding or systematic memory utilization.

**Personalization for embodied agents.** The importance of personalization in robotics has long been recognized (Dautenhahn, 2004; Lee et al., 2012; Clabaugh & Matarić, 2018), particularly in the context of human-robot interaction where robots adapt their interactive behaviors to align with individual users. Recent works have focused on reflecting individual user's preferences during embodied agents' task execution, such as spatial arrangement (Kapelyukh & Johns, 2021; Wu et al., 2023), table settings (Puig et al., 2021), or personalized object navigations (Dai et al., 2024; Barsellotti et al., 2024). Recently, Xu et al. (2024) aim to infer user preferences from a few demonstrations and adapt planning behavior accordingly. However, these approaches primarily address implicit preference adaptation or short-term reactive behaviors derived from repeated user demonstrations, without leveraging explicitly provided, user-defined knowledge in a structured manner.

## 3 EVALUATION SETUP

### 3.1 TASK & FOCUS

**Task: personalized object rearrangement.** We formulate the personalized object rearrangement task as a Partially Observable Markov Decision Process (POMDP) $(S, A, T, R, \Omega, O, \gamma)$, where $S$ denotes the set of environment states, $A$ is the set of actions, $T$ is the transition function, $R$ is the reward function, $\Omega$ is the observation space, $O$ is the observation function, and $\gamma$ the discount factor. At timestep $t$, the agent receives a text observation $w_t \in \Omega$ describing nearby objects based on the previous action $a_{t-1}$ and generates action $a_t$ via LLM policy $\pi$. Given a natural language instruction $I$ that may contain personalized references (e.g., "Place my favorite mug on the table"), the agent must derive the goal representation $g$ using both the instruction and personalized knowledge from memory: $\phi(I, M) \longrightarrow g = (o_i, l_i)_{i=1}^{k}$, where $M = h_1, h_2, \ldots, h_n$ represents the set of episodic memories from previous interactions, and each $(o_i, l_i)$ pair specifies target object $o_i$ and location $l_i$. The policy generates actions based on the observation-action trajectory $\tau_t$:

$$\pi(I, M, \tau_t) \to a_t, \quad \tau_t = (w_1, a_1, w_2, a_2, \ldots, w_{t-1}, a_{t-1}, w_t) \quad (1)$$

The goal is to produce action sequence $a_{1:t}$ such that the resulting state $s_t$ satisfies all object-location specifications in $g$.

**Focus: memory utilization.** Personalized assistance tasks require agents to integrate multiple capabilities including natural language understanding, memory retrieval, planning, and action execution. Among these interconnected abilities, we focus specifically on memory utilization—the agent's capacity to effectively recall and apply relevant personalized knowledge from episodic memory $M$ to interpret instructions and guide task planning. This focus is motivated by memory utilization being the primary distinguishing capability from an agent perspective that separates personalized assistance tasks from conventional embodied tasks, making it essential to investigate for potential challenges and solutions for developing personalized embodied agents.

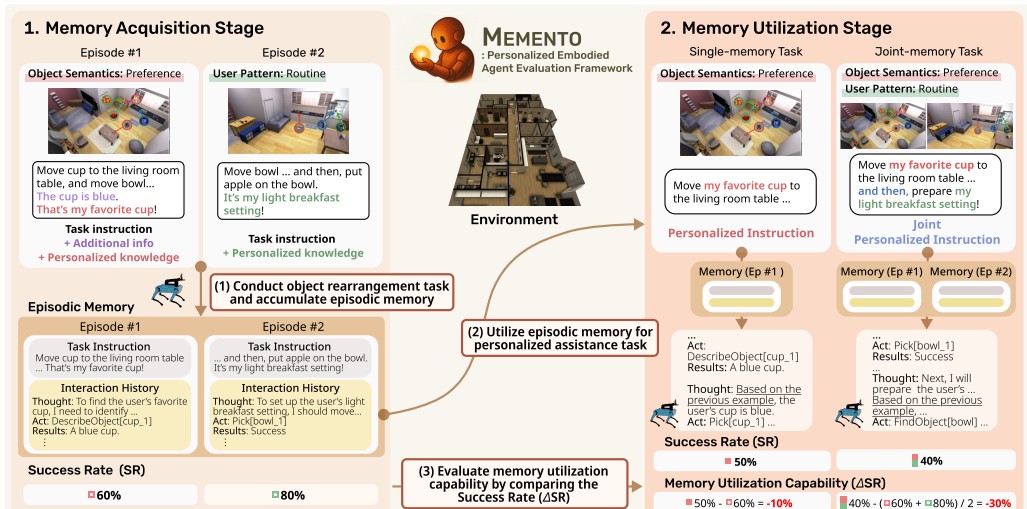

Figure 2: Overview of MEMENTO. The framework evaluates memory utilization capability by comparing agent performance on tasks with identical goals but varying instructions on each stage.

## 3.2 DIMENSIONS OF PERSONALIZED KNOWLEDGE ANALYSIS

We analyze embodied agents' memory utilization capabilities along two dimensions that reflect how personalized knowledge affects goal interpretation in $\phi(I, M) \rightarrow g = (o_i, l_i)_{i=1}^{k}$. Therefore, we categorize personalized knowledge into two types, each designed to isolate distinct reasoning challenges that agents must resolve using episodic memory:

- **Object semantics** evaluates agents' ability to identify target objects $o_i$ based on personal meaning assigned by users, including ownership (*my cup*), preference (*my favorite running gear*), personal history (*a graduation gift from my grandma*), or grouped references (*my childhood toy collections*). This dimension tests whether agents can recall specific object semantics from prior interactions to resolve ambiguous object references.

- **User patterns** assesses agents' capacity to reconstruct complete goals $g$ by leveraging consistent behavioral patterns, including personal routines (*my remote work setup*) and arrangement preferences (*my cozy dinner atmosphere*). This dimension evaluates whether agents can apply previously observed patterns across multiple objects and locations to infer task requirements.

Each dimension targets different aspects of the grounding function $\phi(I, M) \rightarrow g$: object semantics focuses on resolving individual object references, while user patterns addresses holistic goal reconstruction from behavioral context. Detailed explanation of the sub-categories of personalized knowledge can be found in Appendix C.2.

## 3.3 MEMENTO: PERSONALIZED EMBODIED AGENT EVALUATION FRAMEWORK

**Framework design.** The major challenge of evaluating embodied agents' memory utilization capability is quantifying the memory effect on overall task performance. To address this, as shown in Figure 2, we design an end-to-end two-stage evaluation process that measures memory utilization capabilities through instruction interpretation. Given the base object rearrangement task episode defined as the tuple $\epsilon = (S, I, g)$, the key concept of our evaluation process is to share the scene $S$ and goal representation $g$, while varying the instruction $I$ across the two stages to isolate instruction interpretation capability as the primary factor influencing performance differences.

In the **memory acquisition stage**, agents perform conventional object rearrangement tasks with instructions that contain sufficient personalized knowledge. These episodes are defined as $\epsilon_{acq} = (S, I_{acq}, g)$ where $\phi(I_{acq}) \longrightarrow g$, meaning agents can directly infer the goal from instructions. During this stage, agents establish a reference performance baseline while accumulating episodic memory $h_{acq}$ from their interactions. The **memory utilization stage** presents agents with identical tasks

but uses underspecified instructions $\epsilon_{util} = (S, I_{util}, g)$. These tasks can only be completed when agents recall and apply relevant personalized knowledge, requiring the extended grounding function $\phi(I_{util}, h_{acq}) \longrightarrow g$. Through this evaluation process, we are able to quantify the agent's memory utilization capability by comparing performance between the two stages.

Furthermore, to evaluate different complexity levels, we assess both (1) **single-memory tasks** requiring information from one episodic memory, and (2) **joint-memory tasks** necessitating synthesis from two distinct memories, formulated as $\epsilon_{util}^{joint} = (S, I_{util}^{joint}, [g^i; g^j])$ where $i, j$ denote corresponding reference episodes from the acquisition stage. As shown in Figure 3, we conduct experiments without memory usage to validate our framework design, confirming that agents struggle significantly with underspecified instructions.

**Dataset construction.** We construct the dataset for ME-MENTO using Habitat 3.0 simulator (Puig et al., 2023) with a simulated Spot robot (Boston Dynamics, 2025; Yokoyama et al., 2023). Our dataset spans 12 scenes with a total of 438 episodes, using PartNR (Chang et al., 2024) test set as foundation data since its multi-object rearrangement tasks align with our evaluation design.

Using GPT-4o, we generate personalized knowledge and create two instruction types: $I_{acq}$ concatenating base instruction, visual captions, and personalized knowledge for direct goal inference; and $I_{util}$ containing personalized references requiring memory recall. To prevent agents from identifying targets without personalized knowledge,

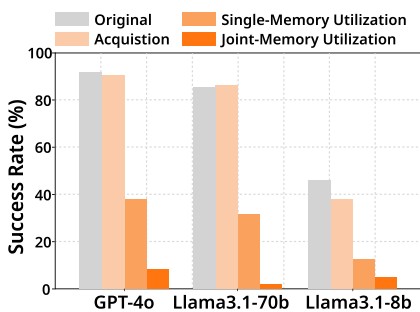

Figure 3: The performance results without using episodic memory. Original indicates episodes from PartNR dataset.

we augmented scenes by placing distractors of the same type but with different appearances near target objects (e.g., placing a "red cup" next to the target "blue cup"). For quality control, we filtered episodes with similar memories referencing identical objects and manually reviewed failed cases to ensure instruction quality. We further validate the dataset with an inter-rater reliability analysis, achieving an average Krippendorff's alpha of 0.69 across four key criteria. Details of the MEMENTO are provided in Appendix C.3.

### 3.4 METRICS & RETRIEVAL SETUP

**Evaluation metrics.** Following Chang et al. (2024), we use two main metrics: **Percent Complete (PC)** for the proportion of goal completion, and **Success Rate (SR)** for full task completion. We also report Sim Steps, which show the number of simulation steps required to complete the task, and Planning Cycles, which indicate the number of LLM inference calls made during task execution. Importantly, we report performance differences between acquisition and utilization stages as $\Delta$**PC** and $\Delta$**SR** to quantify agents' memory utilization capabilities. Note that for joint-memory tasks, these differences are computed relative to the average score of the corresponding acquisition stage episodes.

**Retrieval setup.** We first run the memory acquisition stage for all episodes to accumulate episodic memories, which are then used in the memory utilization stage. For memory retrieval in the utilization stage, we use top-$k$ similarity-based retrieval with *all-mpnet-base-v2* (Reimers & Gurevych, 2019), treating the current instruction as a query and episodic memory instructions as keys. Since our primary objective is evaluating memory utilization capability rather than retrieval performance, we ensure the corresponding gold memories are always included in the top-$k$ results through randomly replacing retrieved candidates when necessary. We set $k = 5$ as our default configuration because this achieves 96.5% recall for gold memories in realistic retrieval scenarios, compared to only 86.1% recall at $k = 3$. Comprehensive analysis of various retrieval strategies without gold memory guarantees is provided in Appendix E.3.

### 4 EVALUATING MEMORY UTILIZATION ACROSS EMBODIED AGENTS

Through MEMENTO, we assess memory utilization capabilities across multiple LLM-powered embodied agents of varying parameter sizes to understand how effectively they provide personalized assistance and identify their limitations (RQ1).

Table 1: Model performance across memory acquisition and utilization stage in MEMENTO.

| Model | Stage | Task Type (Memory) | Planning Cycles ↓ | Sim Steps ↓ | Percent Complete ↑ | △ PC ↑ | Success Rate ↑ | △ SR ↑ |
|---|---|---|---|---|---|---|---|---|
| GPT-4o | Acquisition | - | 16.5 | 2156.1 | 96.3 | - | 95.0 | - |
| | Utilization | Single | 16.1 | 2450.8 | 88.0 | **-8.3** | 85.1 | **-9.9** |
| | | Joint | 28.9 | 3480.7 | 86.7 | **-10.5** | 63.9 | **-30.5** |
| Claude-3.5-Sonnet | Acquisition | - | 16.0 | 2104.1 | 96.2 | - | 94.0 | - |
| | Utilization | Single | 15.3 | 2258.8 | 69.3 | **-26.9** | 63.7 | **-30.3** |
| | | Joint | 27.8 | 3198.8 | 64.6 | **-30.1** | 33.3 | **-57.0** |
| Qwen-2.5-72b | Acquisition | - | 17.5 | 2281.9 | 93.5 | - | 91.0 | - |
| | Utilization | Single | 17.5 | 2691.2 | 72.6 | **-20.9** | 67.2 | **-23.8** |
| | | Joint | 31.3 | 4027.1 | 68.9 | **-27.9** | 36.1 | **-58.3** |
| Llama-3.1-70b | Acquisition | - | 17.7 | 2162.1 | 92.9 | - | 90.0 | - |
| | Utilization | Single | 19.0 | 2566.6 | 72.2 | **-20.7** | 66.7 | **-23.3** |
| | | Joint | 31.4 | 3425.2 | 51.3 | **-44.9** | 8.3 | **-83.4** |
| Llama-3.1-8b | Acquisition | - | 19.3 | 2377.0 | 78.1 | - | 68.5 | - |
| | Utilization | Single | 19.0 | 3131.7 | 48.1 | **-30.0** | 35.0 | **-33.5** |
| | | Joint | 27.4 | 3478.2 | 35.3 | **-45.5** | 8.3 | **-59.8** |
| Qwen-2.5-7b | Acquisition | - | 21.7 | 2476.8 | 64.1 | - | 53.2 | - |
| | Utilization | Single | 21.8 | 3271.0 | 39.1 | **-25.0** | 27.4 | **-25.8** |
| | | Joint | 26.9 | 4149.0 | 33.7 | **-34.2** | 5.6 | **-52.7** |

## 4.1 MODELS & IMPLEMENTATION DETAILS

We evaluate both proprietary models (GPT-4o (Hurst et al., 2024), Claude-3.5-Sonnet (Anthropic, 2024)) and open-source models (Llama-3.1-70b/8b (Grattafiori et al., 2024), Qwen-2.5-72b/7b (Yang et al., 2024)) with different model families and sizes. We implement LLM-powered embodied agents following established architectures (Szot et al., 2021; Puig et al., 2023; Chang et al., 2024), where LLM serves as a high-level policy planner that selects appropriate skills from a predefined skill library using ReAct (Yao et al., 2023) prompting format. Note that, we use gold perception and motor skills to focus on challenges related to embodied agents' cognitive abilities. Complete implementation details are provided in Appendix C.1.

## 4.2 RESULTS

**LLM-powered embodied agents struggle with understanding personalized knowledge.** In Table 1, we observe that while GPT-4o maintains a relatively high success rate in the single-memory tasks, all models show a success rate drop over 20% compared to the memory acquisition stage. This substantial performance decline across all evaluated models demonstrates that current embodied agents cannot reliably integrate personalized knowledge for task execution.

**Current embodied agents can effectively recall object semantics but struggle to comprehend user patterns.** We then analyze the agents performance on single-memory task by personalized knowledge types. Notably, as shown in Figure 4, all models

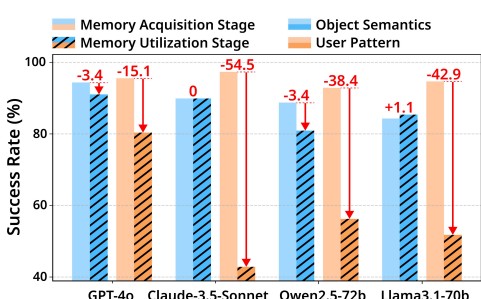

Figure 4: The results of personalized knowledge type based analysis (single-memory).

show minimal performance degradation for object semantics tasks, while user pattern tasks exhibit substantial performance drops across all evaluated models, indicating that current embodied agents face significant limitations in comprehending sequential information within behavioral contexts.

**Agents default to commonsense over personalized knowledge.** To understand the agents' behavior on personalized assistance tasks, we conduct qualitative case study which is shown in Figure 5. Through this analysis, we identify an interesting pattern: *agents tend to rely on commonsense knowledge rather than personalized memory*. For example, when reasoning about "afternoon refresh setup" (C.1), the agent relies on commonsense knowledge rather than referencing personalized memory, correctly predicting "jug and cup" because this commonsense happens to align with the

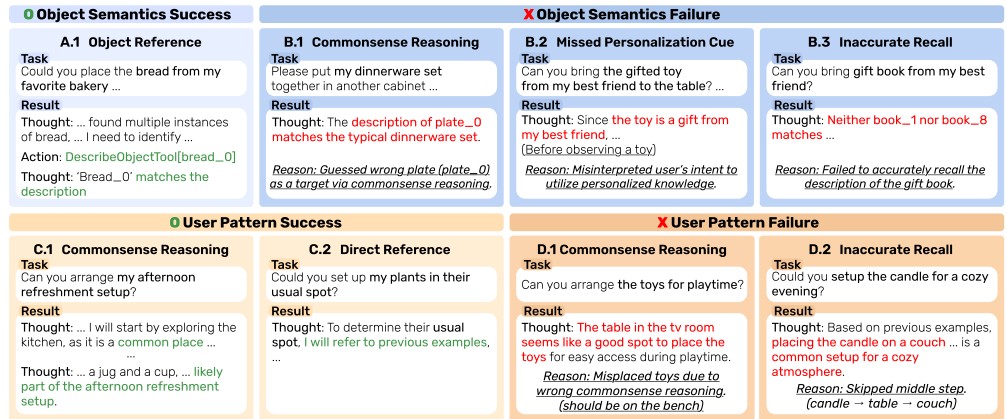

Figure 5: Taxonomy of successful and failed cases in memory utilization with illustrative examples. Top: success and failure cases of object semantics; Bottom: success and failure cases of user patterns.

user's actual behavior. However, when applying the same commonsense-based reasoning approach to "playtime" (D.1), the agent incorrectly places toys on the TV room table instead of consulting the personalized memory that specifies the user's preference for placing toys on the bench. This indicates that when agents struggle to reference personalized knowledge from memory, they default to parametric commonsense knowledge rather than leveraging non-parametric personalized knowledge, leading to inconsistent performance across user contexts.

## 5 SCENARIO-BASED ANALYSIS FOR MEMORY UTILIZATION BOTTLENECKS

In this section, we conduct scenario-based analysis to understand the factors that negatively affect agents' memory utilization capabilities in personalized assistance tasks (RQ2). First, we examine agent performance when increasing the number of retrieved memories in the context (§ 5.1). Second, we analyze agent behavior when tasks require coordinating multiple memories to achieve task completion (§ 5.2).

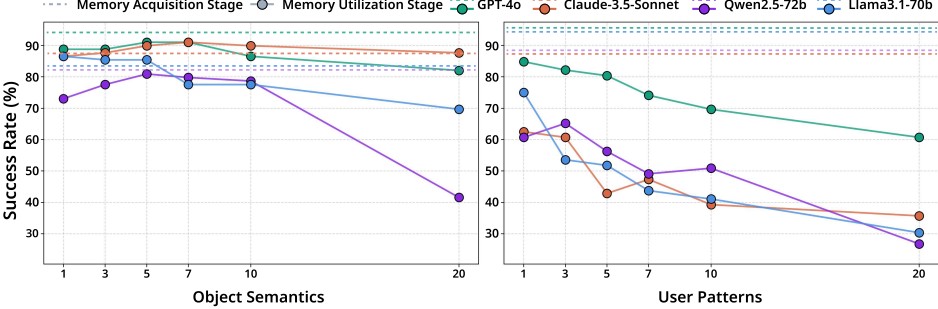

Figure 6: Success rate comparison across models as top-k value increases. Dashed lines represent memory acquisition stage baselines for each model.

### 5.1 ANALYZING AGENT PERFORMANCE UNDER INCREASED MEMORY CONTEXT

To understand whether agents remain robust in identifying correct information as memory volume increases, we vary the number of retrieved memories (top-$k$) in single-memory tasks. We ensure gold memory is always included and focus on whether agents can reference the correct memory.

**Information overload degrades memory utilization performance.** As shown in Figure 6, increasing $k$ consistently degrades performance across all models and both personalized knowledge types, indicating that agents struggle to extract relevant information when presented with larger sets of retrieved memories. We found that agents increasingly rely on commonsense knowledge rather than personalized memories as retrieved memories increase, especially on user patterns, suggesting that embodied agents suffer from information overload. Therefore, even though LLMs demonstrate impressive capabilities in processing long-context information (Wang et al., 2024b; Liu et al., 2025),

which we confirm in Appendix D.5, memory systems that retrieve appropriate numbers of memories are crucial for personalized embodied agents.

## 5.2 INVESTIGATING AGENT BEHAVIOR IN JOINT-MEMORY TASKS

In realistic scenarios, users often refer to multiple pieces of personalized knowledge within a single instruction. Accordingly, we fix the top-$k$ parameter to $k = 5$ and employ joint-memory tasks in MEMENTO to analyze the agents' performance on tasks that require synthesizing information from two distinct episodic memories.

**Agents fail to coordinate multiple memories.** As shown in Table 1, all models exhibit substantially larger performance drops in joint-memory tasks compared with single-memory tasks. Even GPT-4o's success rate declines by 30.5%, revealing serious bottlenecks when agents must coordinate multiple memories for personalized assistance. Figure 7 breaks down the joint-memory results by personalized-knowledge type and, consistent with earlier findings, agents perform particularly poorly on instructions involving user patterns. Notably, unlike in single-memory tasks where the ΔSR for object semantics was only marginal, GPT-4o shows a -20.8% drop when combining two object-semantic memories. Because episodic memory inherently en-

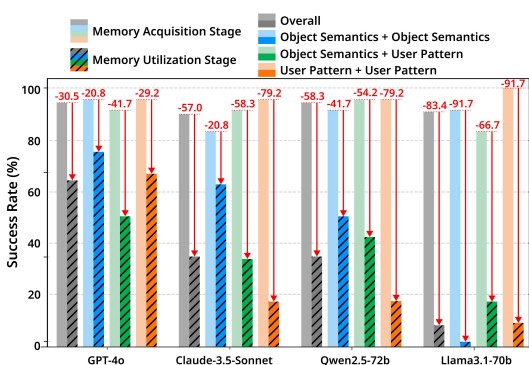

Figure 7: Personalized knowledge type-based analysis on joint-memory tasks.

codes interaction history, the in-context learning nature of LLMs makes it difficult to simultaneously reference multiple distinct memories. These findings underscore that agents struggle to effectively extract personalized knowledge using current memory architectures, highlighting the need for improved memory designs to support personalized assistance tasks.

## 6 EXPLORING MEMORY DESIGN FOR PERSONALIZED EMBODIED AGENTS

Based on our previous findings that demonstrate critical bottlenecks in memory utilization for personalized assistance, we identify that these limitations stem from the fundamental challenge of extracting relevant personalized knowledge from episodic memories. We hypothesize that clearly providing personalized knowledge information—an approach consistent with recent advances in context engineering (Mei et al., 2025)—will enhance agents' ability to leverage memory effectively. Therefore, in this section, we explore memory architectural approaches to improve agents' memory utilization capabilities for personalized assistance tasks (RQ3).

Table 2: Model performance with memory format simplification approaches. Full memory indicates the episodic memory.

| Model | Memory | PC (%) | SR (%) |
|---|---|---|---|
| GPT-4o | Full Memory | 90.0 | 83.3 |
| | Summarization | 88.0 | 83.3 |
| | Instruction-only | 62.4 | 50.0 |
| Qwen-2.5-72b | Full Memory | 77.2 | 66.7 |
| | Summarization | 77.4 | 70.0 |
| | Instruction-only | 51.3 | 40.0 |
| Llama-3.1-8b | Full Memory | 72.8 | 63.3 |
| | Summarization | 49.4 | 43.3 |
| | Instruction-only | 40.0 | 30.0 |
| Qwen-2.5-7b | Full Memory | 50.1 | 43.3 |
| | Summarization | 43.9 | 36.7 |
| | Instruction-only | 35.6 | 23.3 |

### 6.1 INVESTIGATING MEMORY FORMAT SIMPLIFICATION

As episodic memory contains full interaction histories, it includes contextual information irrelevant to personalized knowledge extraction. Therefore, we investigate memory format simplification approaches that provide agents with refined information by removing irrelevant content, hypothesizing that this reduction of noise will improve performance. We compare full episodic memory against two simplified alternatives: (1) GPT-4o summarized episodic memory that retains key information while removing trajectory details, and (2) instruction-only setups that rely solely on task instructions, since personalized knowledge can be extracted from instructions alone.

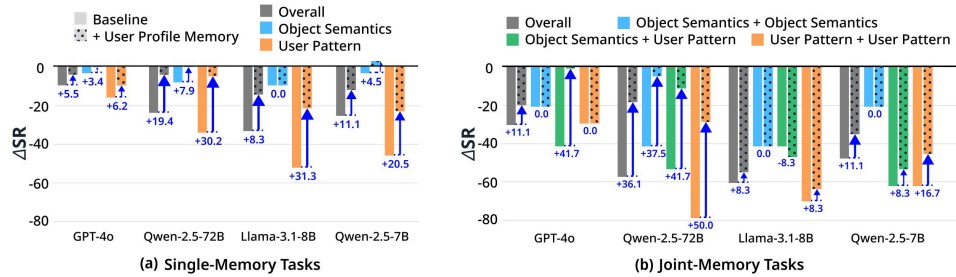

Figure 8: Agent performance with user profile memory across single- and joint-memory tasks.

**Episodic memory provides both personalized knowledge and in-context learning benefits.** As shown in Table 2, summarized memory shows minimal impact on large models (GPT-4o, Qwen-2.5-72b). Notably, smaller models (Llama-3.1-8b, Qwen-2.5-7b) exhibit performance degradation when using summarized memory compared to full episodic memory. This performance drop occurs because summarizing removes the trajectory information that provides crucial in-context learning benefits for LLMs, reducing the overall task success rate. These findings demonstrate that episodic memory is an essential module for embodied agents.

## 6.2 DEVELOPING HIERARCHICAL KNOWLEDGE GRAPH-BASED USER PROFILE MEMORY

Building upon our finding that episodic memory is an essential memory module, we devise a hierarchical knowledge graph-based *user profile memory* module that manages personalized knowledge independently to provide cleaner, more accessible personalized information to the agent.

**Memory module design.** A key challenge in developing a personalized knowledge module is organizing knowledge relationships effectively. Sequential knowledge like user patterns requires sophisticated memory structure to prevent information loss or failing to adapt to evolving knowledge (*e.g.*, "Add the mug from my coffee set to my morning routine and prepare my routine"). Inspired by previous works on knowledge graphs (Speer et al., 2017; Bosselut et al., 2019), we construct personalized knowledge as a hierarchical knowledge graph that captures both object semantics and user patterns.

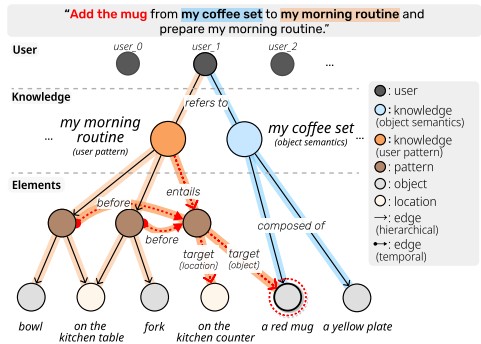

Figure 9: Illustration of user profile memory.

As shown in Figure 9, user profile memory employs a three-level structure with users at the top, knowledge types (object semantics and user patterns) in the middle, and specific elements (objects, patterns, locations) at the bottom. The graph connects these levels using hierarchical edges for structural relationships and temporal edges for sequential ordering within user patterns. This design enables systematic representation and easy updates of personalized knowledge—for instance, a red mug from a coffee set can be dynamically added to a morning routine by updating the relevant knowledge relationships. More details of the user profile memory design, schema, extraction and retrieval algorithms are provided in Appendix D.1.

**User profile memory enhances embodied agents' memory utilization capability.** As shown in Figure 8, agents with user profile memory achieves significant performance improvements across all models for both single-memory and joint-memory tasks. The performance gains are especially notable on user pattern tasks, where the sequential structure had previously posed challenges. These results indicate that user profile memory effectively manages personalized knowledge, demonstrating its ability to enhance agents' memory utilization capabilities for personalized assistance tasks.

## 7 CONCLUSION

This work investigates memory utilization challenges in LLM-powered embodied agents for personalized assistance through MEMENTO, revealing that while agents can recall simple object semantics,

they struggle with sequential user patterns, suffer from information overload when processing multiple retrieved memories, and fail to coordinate joint-memory tasks effectively. We demonstrate that episodic memory provides both personalized knowledge and in-context learning benefits, explaining why simple summarization approaches fail. To address these bottlenecks, we design a hierarchical knowledge graph-based user-profile memory module that separately manages personalized knowledge while preserving episodic memory benefits, achieving substantial improvements across both single-memory and joint-memory tasks. We hope these findings will inspire future research toward more sophisticated memory architectures that enable truly effective personalized embodied agents.

## ACKNOWLEDGEMENTS

This project is supported by Microsoft Research Asia. This research was supported by Institute of Information & communications Technology Planning & Evaluation (IITP) grant funded by the Korea government (MSIT), under the Global Research Support Program in the Digital Field program (RS-2024-00436680), National AI Research Lab Project (RS-2024-00457882), Artificial Intelligence Graduate School Program (Yonsei University) (RS-2020-II201361). We also thank Donghyun Kim, Juhyun Park, and Hyeonjong Ju for their helpful discussions and support. Jinyoung Yeo is the corresponding author.

## ETHICS STATEMENT

This work involves no human subjects, conflicts of interest, or legal compliance concerns. We use licensed resources (*e.g.*, PARTNR (Chang et al., 2024) under the MIT License) and commonsense knowledge without personal data, copyrighted content, or bias-amplifying material. We generated personalized knowledge from LLM-synthesized descriptions with prompts designed to reduce biased or discriminatory content, and we manually verified that the resulting user knowledge was free of such concerns. We acknowledge that embodied memory systems may pose privacy and security risks in real-world deployment. To mitigate these issues, future systems should adopt on-device processing, minimize retention, require user consent, and apply secure, transparent memory handling.

## REPRODUCIBILITY STATEMENT

To facilitate full reproducibility of our results, we include complete evaluation dataset details and experimental procedures in the appendices. Furthermore, we publish our code at `https://github.com/Connoriginal/MEMENTO` with thorough documentation.

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

# A Limitations, Privacy, Societal Impacts and LLM Usage

## A.1 Limitations

This work focuses on agents' memory utilization capabilities for providing personalized assistance. While our study provides valuable insights, several limitations must be acknowledged. First, our experiments were conducted in a controlled simulator environment (Puig et al., 2023). Although we focused on agents' cognitive abilities and expect marginal impact on our results, this controlled environment may not fully capture the complexity and variability of real-world, presenting potential risks for generalization. Second, we utilize oracle perception and motor skills to isolate memory utilization as the primary performance factor. Recent research has increasingly incorporated visual components into embodied agents (Huang et al., 2023a; Yang et al., 2025) or developed Vision-Language-Action (VLA) models that generate end-to-end actions (Brohan et al., 2023; Kim et al., 2024; Bjorck et al., 2025). Additionally, recent studies have shown that primary limitations in embodied tasks often exist at the observation and action levels (Chang et al., 2024; Sapkota et al., 2025). Therefore, while our focus on language-based memory utilization capabilities provides clear insights into this specific aspect, it may not capture all failure modes present in actual embodied agents. Finally, our framework lacks conversational components that are crucial for effective personalized assistance. Beyond memory utilization, personalized embodied agents require robust communication capabilities, including the ability to clarify ambiguous instructions, proactively engage with users, and adapt their communication style to individual preferences. Investigating how memory utilization integrates with conversational abilities to enable more comprehensive personalized assistance represents an important direction for our ongoing research.

## A.2 Privacy and Security Considerations

As our work involves personalized knowledge and agent memory systems, we recognize potential privacy risks including unauthorized access to user-specific knowledge and inference of sensitive personal information from interaction patterns. While our current study utilizes synthetic personalized data to mitigate privacy concerns, real-world deployment of personalized embodied agents requires robust privacy safeguards including encryption of user-specific knowledge bases, secure access

controls with proper authentication protocols, data minimization strategies, and mechanisms for user control over personal data. The specific implementation of these privacy protection mechanisms should depend on deployment context and applicable regulatory requirements, and we recommend thorough privacy impact assessments before real-world deployment.

### A.3 SOCIETAL IMPACTS

Our work explores how embodied agents can remember and adapt to user-specific preferences through episodic memory, enabling more personalized and natural interactions. This capability has the potential to enhance convenience, efficiency, and user satisfaction in everyday environments—benefiting a wide range of users regardless of age or ability. However, personalization also raises concerns about privacy, bias reinforcement, and over-dependence on AI agents (Rueben & Smart, 2016; Lee et al., 2012). Since episodic memory involves storing interaction history, future systems must consider secure and transparent memory handling. Although our study is conducted in a controlled simulator, these considerations will become crucial as such systems move toward real-world deployment. We hope that MEMENTO serves as a foundation for future research in building safe, privacy-aware, and user-aligned personalized embodied agents.

### A.4 LLM USAGE

Our use of LLMs was for the following specific purposes:

- **Writing and Editing Assistance:** We utilize LLMs for improving the clarity, grammar, and readability of the text. This included correcting grammatical errors, and ensuring consistent terminology throughout the paper.
- **Code Generation for Experiments:** We utilize LLMs to accelerate the implementation of our experimental framework.

The authors meticulously reviewed, edited, and validated all content generated or modified by LLMs. The conceptual arguments, experimental design, analysis, and final conclusions presented in this paper are the original work of the authors. We take full responsibility for all content of this work.

## B EXTENDED RELATED WORK

Our work focuses on memory and personalization in embodied agents. Here, we broaden the related work to include studies on memory-augmented agents, personalization, and memory architectures within the wider research context.

**Memory-augmented agents** have increasingly been studied as a means to support long-term reasoning, planning, and adaptive behavior in LLM agents. Park et al. (2023) propose *Generative Agents*, which simulate human-like behavior by maintaining a memory stream of past experiences in natural language. This memory stream enables agents to reflect on prior events, retrieve relevant information, and plan future actions based on their histories. In a similar direction, Xu et al. (2025) introduce *A-Mem*, a dynamic memory system in which memory is represented as evolving and interconnected notes. The agent can generate, retrieve, and update these notes over time, thereby supporting adaptability.

**Memory for dialogue and personalization** has emerged as a key direction, highlighting how memory supports user-specific context and long-term consistency in interactions. Li et al. (2025a) present a system that leverages both short-term and long-term memories to maintain user-specific context across multiple conversation sessions, reporting improvements in dialogue coherence and persona consistency. Meanwhile, Wang et al. (2024a) propose the Self-Controlled Memory (SCM) framework, where the agent dynamically decides when and what to store or retrieve, achieving greater coherence and knowledge retention over extended interactions. Alongside these approaches, structured long-term and episodic memory have been introduced to enhance personalization. Zhong et al. (2023) develop *MemoryBank*, which stores and retrieves interaction history to support consistent responses across turns. Das et al. (2024) present *Larimar*, a framework that enables selective recall and forgetting, providing controllable knowledge updates that can support personalization at scale.

**Architectural perspectives on memory** focus not only on building memory components for individual tasks but on treating memory as a fundamental part of the overall system. Packer et al. (2024) illustrate this idea with *MemGPT*, which frames memory management through an operating-system analogy. In this view, the LLM functions like an operating system that autonomously organizes and manages both its in-context and external memory, rather than relying solely on user-provided prompts. This perspective emphasizes memory as a central architectural layer, enabling scalable and autonomous agents that can handle complex, long-horizon tasks.

## C  DETAILS OF EVALUATION SETUP

### C.1  LLM-POWERED EMBODIED AGENT ARCHITECTURE

Following Szot et al. (2021); Puig et al. (2023); Chang et al. (2024), we adopt a two-layer hierarchical control architecture for our LLM-powered embodied agent. We utilize the LLM as a high-level policy planner that selects appropriate skills from the predefined skill library. The selected skill then provides control signals to the simulator. For memory systems, we implement a textual scene-graph as our semantic memory alongside an episodic memory.

**Skill library.**    The skill library consists of oracle low-level skills that the LLM policy can select as actions. These action skills are divided into motor skills (*e.g.*, `navigate`, `pick`, `place`) and perception skills (*e.g.*, `describe_object`, `find_object`, `find_receptacle`). Note that we exclusively used oracle skills in our skill library to focus on episodic memory-centric analysis. Further descriptions are provided in Table 3 and Table 4

Table 3: List of available agent motor skills.

| Skill | Description |
|---|---|
| Navigate | Used for navigating to an entity. You must provide the name of the entity you want to navigate to. Example: Navigate[counter_22]. |
| Pick | Used for picking up an object. You must provide the name of the object. Example: Pick[cup_1]. |
| Place | Used for placing an object on a target location. Example: Place[book_0, on, table_2, None, None]. |
| Open | Used for opening an articulated entity. Example: Open[chest_of_drawers_1]. |
| Close | Used for closing an articulated entity. Example: Close[chest_of_drawers_1]. |
| Explore | Doing exploration towards a target object or receptacle, you need to provide the name of the place you want to explore. |
| Wait | Used to make agent stay idle for some time. Example (Wait[]) |

Table 4: List of available agent perception skills.

| Skill | Description |
|---|---|
| FindObjectTool | Used to find the exact name/names of the object/objects of interest. If you want to find the exact names of objects on specific receptacles or furnitures, please include that in the query. Example (FindObjectTool[toys on the floor] or FindObjectTool[apples]). |
| FindReceptacleTool | Used to know the exact name of a receptacle. A receptacle is a furniture or entity (like a chair, table, bed etc.) where you can place an object. Example (FindReceptacleTool[a kitchen counter]). |
| FindRoomTool | Used to know the exact name of a room in the house. A room is a region in the house where furniture is placed. Example (FindRoomTool[a room which might have something to eat]). |
| DescribeObjectTool | Used to retrieve a brief descriptive or semantic meaning of a given object or furniture name. Example (DescribeObjectTool[sponge_1]). |

**Semantic memory.**    For semantic memory, we implement a scene-graph style hierarchical representation, which has demonstrated effectiveness for planning problems (Agia et al., 2022; Rana et al., 2023; Gu et al., 2024). Following Chang et al. (2024), we utilize a multi-edge directed graph with three distinct levels to represent environmental entities. The top level contains a single root

node representing the house environment, the second level comprises room nodes, and the third level encompasses furniture, objects, and agents. Each node stores the corresponding entity's 3D location and relevant state information. This graph structure is initialized and continuously updated with ground-truth information from the simulator at each state $s_t$. In our system, this structured semantic memory provides the LLM planner with an interpretable representation of the environment, which can be flexibly queried and reasoned over through natural language descriptions.

**Episodic memory.** Our episodic memory is configured to store the ReAct-style formatting that guides the LLMs' reasoning process (Yao et al., 2023). This memory structure captures both the user's instruction and the complete sequence of `<Thought, Action, Observation>` triplets generated during task execution. The episodic memory is accessed at the beginning of each task by retrieval and updated upon task completion, enabling the agent to recall previous interactions.

**Memory retrieval.** For retrieval, we encode instructions and memory entries using the *all-mpnet-base-v2* sentence transformer (Reimers & Gurevych, 2019) and use the current task instruction as the query. To avoid ambiguous or irrelevant memories, we retrieve candidate memories only from the history within the same scene as the current task.

**LLM inference setup.** We configured the language model with a temperature of 0 to ensure deterministic outputs. For sampling parameters, we set `top_p` to 1 and `top_k` to 50.

## C.2 PERSONALIZED KNOWLEDGE DETAILS

We categorize knowledge about personal items as *object semantics* and knowledge about consistent behaviors as *user patterns* to structure our evaluation approach. *Object semantics* can be further classified into four sub-categories: naive ownership (*e.g.*, "my cup"), object preference (*e.g.*, "a chessboard I play chess with my brother"), history (*e.g.*, "graduation gift from my grandma"), and group (*e.g.*, "my favorite toys", where toys indicate toy airplane and toy truck). *User patterns* encompass consistent action sequences in specific contexts, with two sub-categories: personal routine (*e.g.*, "my remote work setup") and arrangement preference (*e.g.*, "my movie night setup"). Based on these personalized knowledge categories, we designed tasks that specifically require agents to recall and apply this information to evaluate their memory utilization capabilities. Further details are provided in Table 5 and Table 6

Table 5: List of the subcategories for object semantics.

| Type | Description | Example |
|------|-------------|---------|
| Ownership | Possessive reference to the user's belonging | My cup, My laptop |
| Preference | Object aligned with the user's individual taste or selection | Bread from my favorite bakery, Jug for serving drink |
| History | Object linked to personal memory or past experience | photo of the my beloved pet, travel souvenir vase |
| Groups | Conceptual or functional grouping of multiple related objects | my home office setup, my travel essentials |

Table 6: List of the subcategories for user patterns.

| Type | Description | Example |
|------|-------------|---------|
| Routine | A sequence or setup the user follows as a habit or regular activity. | meal time setting, setup for cooking routine |
| Preference | A specific way the user prefers to prepare or arrange their environment when a particular situation occurs. | my coffee break, cozy decoration spot |

### C.3 DETAILS OF MEMENTO

#### C.3.1 FOUNDATION DATASET

We adopt the test set of PartNR (Chang et al., 2024) as our foundation object rearrangement task dataset. PartNR is designed to evaluate planning and reasoning capabilities in embodied tasks and includes four primary task types: (1) constraint-free basic rearrangement, (2) spatial tasks requiring reasoning about object positions, (3) temporal tasks with sequential dependencies, and (4) heterogeneous tasks involving human-only actions. We select PartNR episodes for their complexity beyond simple pick-and-place operations, filtering out tasks irrelevant to personalized assistance scenarios. The rich linguistic structure and diverse task requirements make PartNR particularly suitable for evaluating personalized user patterns, enabling us to create scenarios that effectively test agents' ability to adapt to user-specific behavioral routines.

#### C.3.2 DATASET GENERATION PROCESS DETAILS

We leverage GPT-4o and a systematic process to incorporate personalized knowledge into existing task structures, creating tasks tailored for each evaluation stage.

**Captioning process.** Since we employ LLMs as high-level policy planners for embodied agents, we generate natural language descriptions of objects in the scene using GPT-4o, enabling agents to reason over object properties without relying on direct visual perception. We collect object models from the OVMM dataset (Yenamandra et al., 2023) and use GPT-4o to generate descriptions from rendered object images. In particular, the Google Scanned dataset (Downs et al., 2022) included within OVMM provides object identities in file names, which we leverage alongside the images to produce more realistic and accurate descriptions. Through this process, we generate 1,920 object descriptions spanning 66 object categories. The prompts used for generating object descriptions are provided in Appendix F.

**Preprocessing scenes.** To collect suitable episodes for personalized assistance tasks, we preprocessed and filtered episodes using three criteria. We only gathered non-heterogeneous episodes that can be executed by robotic agents, excluding tasks that require human-only actions. We filtered out tasks where target objects were not uniquely specified (e.g., "Bring one apple to the kitchen table"), since our task setting requires identification of specific objects based on personalized knowledge that distinguishes them from similar objects. When captions for target objects were unavailable, we substituted alternative objects from the same category, or excluded the entire episode if no suitable alternatives existed.

**Distractor sampling.** To sample distractor objects for episodes focusing on object semantics, we utilized PartNR's dataset generation methods. This approach systematically selects objects located adjacent to target objects on the same receptacles or floor surfaces, requiring agents to differentiate between objects of the same category.

**Task instruction generation process.** We prompt GPT-4o to generate personalized knowledge tasks based on the knowledge categories defined in Section C.2.

For object semantics tasks, we provide target object captions and guide GPT-4o to generate natural semantic descriptions. We first instruct the model to create the most plausible subcategory of object semantics, then generate corresponding personalized knowledge. The resulting instruction comprises three components: the command instruction, additional object information, and personalized knowledge. For example: "Bring the cup on the kitchen table" (command) + "The cup is a white mug with fancy handle" (description) + "That mug is my coffee mug" (personalized knowledge).

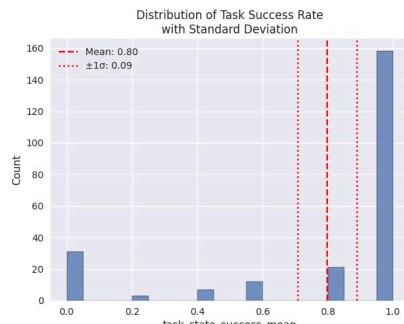

Figure 10: Episodes with zero success rate (31 in total) were excluded from the analysis.

For user pattern tasks, we provide base instructions and allow GPT-4o to infer plausible user patterns corresponding to action sequences. Memory acquisition stage instructions concatenate the command with personalized knowledge (e.g., "Bring the cup and dish from the kitchen table to living room. That's my dinner setup"). Detailed prompts are provided in Appendix F.

**Quality control.** To ensure dataset quality, we implement a two-stage filtering process after generating episodes for MEMENTO. We first apply rule-based heuristic filters guided by six explicit criteria to identify problematic episodes:

- **Semantic misalignment** – Instructions inconsistent with stored memory (e.g., memory states "my favorite cup" but instruction refers to "my cup")
- **Triviality** – Instructions solvable without memory access (e.g., "Prepare my coffee time cup setup on the kitchen table" when setup details are explicitly provided)
- **Redundancy** – Instructions repeating knowledge from previous episodes in the same scene (e.g., "my usual morning cup" when an earlier episode already established "the cup I usually use in the morning")
- **Environment inconsistency** – Instructions conflicting with actual object properties (e.g., describing target cup as red when it's actually blue, but a distractor cup is red)
- **Instruction inconsistency** – Semantic inconsistency between acquisition and utilization stages (e.g., changing from "pick from A and place at B" to only "place at B")
- **Temporal irrelevance** – Instructions with expired temporal context (e.g., referencing "today's temperature" when the memory is no longer current)

Following this rule-based filtering, we test each remaining episode with GPT-4o, excluding episodes where the model fails five consecutive attempts as this indicates potential feasibility or clarity issues. This comprehensive quality control process filters out 31 episodes (13.4%) showing consistently poor performance, all with zero success rates indicating complete task failure. The remaining 201 episodes demonstrate strong performance with 95% confidence intervals of [0.755, 0.844] for success rate and [0.823, 0.893] for completion rate. The high correlation (r = 0.908) between these metrics confirms consistent performance across evaluation measures.

### C.3.3 INTER-RATER RELIABILITY ANALYSIS

To validate the consistency of our heuristic filtering process, we conduct inter-rater reliability analysis using Krippendorff's alpha (Krippendorff, 2011). Multiple annotators independently evaluate a held-out subset of instructions using our six filtering criteria. Since Environment inconsistency and Temporal irrelevance are strictly controlled during instruction generation, they achieve near-perfect agreement and we exclude them from reliability analysis. We compute Krippendorff's alpha for the remaining four categories, obtaining values of 0.75 for Semantic misalignment, 0.87 for Instruction inconsistency, 0.58 for Triviality, and 0.57 for Redundancy. The average alpha of 0.69 across these criteria indicates moderate agreement, supporting the reliability of our filtering process for maintaining dataset quality.

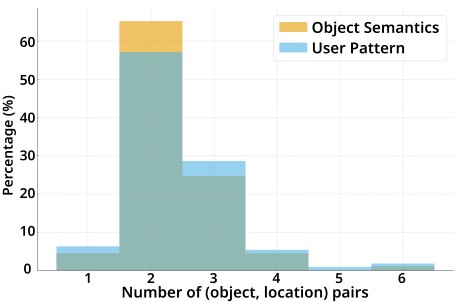

Figure 11: Distribution of required (object, location) pairs across object semantics and user pattern tasks.

### C.3.4 DATASET STATISTICS

The dataset comprises 438 episodes, divided into two main stages. The memory acquisition stage contains 201 episodes (89 object semantics tasks and 112 user patterns tasks). The memory utilization stage also contains 201 single-memory episodes (89 object semantics and 112 user patterns tasks), along with 36 multi-memory episodes, which include 12 object semantics pairs, 12 user patterns pairs, and 12 mixed pairs. All episodes were constructed using the Habitat 3.0 simulator.

**Number of (target, location) pairs.** We compared the number of (object, location) pairs required for successful completion across both object semantics and user pattern tasks. As shown in Figure 11, the distribution of required (object, location) pairs is nearly identical between the two personalized knowledge types. This indicates that the performance difference between object semantics and user patterns stems from the intrinsic characteristics of each knowledge type rather than differences in task complexity measured by the number of required object-location pairs.

## C.4 EVALUATION METHOD

We adopt the official evaluation protocol from the PartNR benchmark (Chang et al., 2024), which provides a python-based framework for assessing multi-step rearrangement tasks. We use this framework without modification. The evaluator analyzes simulator states using three components: (1) **propositions** that define object relationships to satisfy (e.g., `is_on_top([spoon_1], [table_1])`), (2) **dependencies** that define temporal conditions for multi-step instructions (`after_satisfied`, `after_unsatisfied`), and (3) **constraints** that enforce execution requirements (e.g., ordering, object consistency across steps). We rely on this system to evaluate tasks involving ambiguous references and sequential dependencies. The evaluation produces a percent-complete score and binary success indicator.

## C.5 COMPUTING RESOURCES

Our experiments primarily utilized commercial API services rather than local computing resources. We used OpenAI's Chat API for accessing GPT-4o, Claude models through Anthropic's API, and OpenRouter's API service for accessing Llama-3.1 (Meta, 2024), Qwen-2.5 (Yang et al., 2024). For running simulation environment we used 8 NVIDIA GeForce RTX 3090 GPUs. For our implementation and evaluation, we use Huggingface library2, vLLM library. Both libraries are licensed under Apache License, Version 2.0. And we used langchain library, under MIT License. We have confirmed that all of the artifacts used in this paper are available for non-commercial scientific use.

# D EXPERIMENT & ANALYSIS DETAILS

In this section, we provide a detailed description of the experimental setup, analysis results, and the memory module, complementing the explanations provided in the main text.

## D.1 USER PROFILE MEMORY

Our work introduces a *user profile memory* module and offers initial evidence that, when combined with episodic memory, structured personalized knowledge can enhance performance on personalized assistance tasks.

**Knowledge graph design.** We define the user profile memory as a hierarchical knowledge graph that organizes personalized knowledge into three levels—*User*, *Knowledge*, and *Elements*—to provide a structured representation that is both scalable across users and resistant to redundancy from overlapping entities. The hierarchical partitioning enables the system to scale to many users while preserving efficiency, since overlapping objects can be represented in a canonical form and connected via edges instead of being redundantly replicated. To support this representation, each nodes are connected through two types of edges: *hierarchical edges*, which capture inclusion relations across and within levels, and *temporal edges*, which encode the sequential structure of user patterns. Together, these edge types allow the graph to comprehensively represent the personalized knowledge that users may require an embodied agent to leverage in real-world interactions.

**Knowledge graph update procedure.** While the hierarchical design specifies how knowledge is organized, the graph must also evolve as users provide new information or refine existing entries. Algorithm 1 presents the procedure for updating the knowledge graph. Given a user instruction $I$ and the current graph $G$, the LLM processes the instruction by extracting candidate knowledge $\mathcal{K}$ and associated elements $\mathcal{E}$. For each candidate knowledge $k$, the relevant subgraph $G_{t_k}$ is filtered, and similarity search is applied to collect matching nodes $C$. These candidates are then expanded by collecting all connected edges and their descendant nodes, and the resulting subgraphs are passed to an LLM. The LLM determines whether the instruction corresponds to an addition or an update,

executes the corresponding operation. For an update, the LLM removes the existing knowledge node and generates a new one, and for an addition it simply creates a new knowledge node. In both cases, the LLM determines whether to reuse or instantiate the associated object, location nodes based on whether their similarity scores exceed a predefined threshold.

---

**Algorithm 1:** Update Knowledge Graph

---

**Input:** User instruction $I$, Knowledge graph $G$
**Output:** Updated knowledge graph $G'$
```
// Step 1:  Extract candidate knowledges from instruction
```
$\mathcal{K} \leftarrow ExtractKnowledge(I)$ // $\mathcal{K} = \{(k_1, t_1), (k_2, t_2), \dots\}$
$C \leftarrow \varnothing$

**foreach** $(k, t_k) \in \mathcal{K}$ **do**
```
   // Step 2:  Extract elements (patterns/objects/locations)
      tied to this knowledge
```
 $\mathcal{E} \leftarrow ExtractElements(I, k, t_k)$;
```
   // Step 3:  Select subgraph by type
```
 $G_{t_k} \leftarrow Subgraph(G, type = t_k)$;
```
   // Step 4:  Retrieve candidates for knowledge and elements
```
 $C \leftarrow C \cup SimilaritySearch(G_{t_k}, k, t_k)$;
 **foreach** $(e, t_e) \in \mathcal{E}$ **do**
  $C \leftarrow C \cup SimilaritySearch(G_{t_k}, e, t_e)$;

```
// Step 5:  Decide add or update
```
$op \leftarrow DecideAddOrUpdate(I, C)$;

**if** $op = $ update **then**
 **return** $G' \leftarrow UpdateKnowledgeNode(G, Expand(C), I)$;
**else if** $op = $ add **then**
 **return** $G' \leftarrow AddKnowledgeNode(G, Expand(C), I)$;

---

**Algorithm 2:** RetrievalProcedure

---

**Input:** User instruction $I$, User profile memory graph $G$
**Output:** Retrieved natural language descriptions $R$
```
// Step 1:  Parse instruction
```
$\mathcal{K} \leftarrow ExtractKnowledge(I)$;
$\mathcal{C}, \mathcal{R} \leftarrow \varnothing$
```
// Step 2:  Filter subgraph and run similarity search
```
**foreach** $(k, t_k) \in \mathcal{K}$ **do**
 $G_t \leftarrow Subgraph(G, type = t)$;
 $C \leftarrow C \cup SimilaritySearch(G_{t_k}, k, t_k)$;
$C \leftarrow RemoveDuplicates(C)$;
```
// Step 3:  Reformulate results with LLM
```
**foreach** $(c, t_c) \in \mathcal{C}$ **do**
 $R \leftarrow R \cup Reformulate(Expand(c))$;
**return** $R$

---

**Retrieval procedure.** To leverage personalized knowledge during task execution, the agent retrieves information from the user profile memory using a hybrid retrieval pipeline that combines embedding-based search with LLM reasoning. This procedure ensures that the retrieved knowledge is both semantically aligned with user instructions and interpretable in natural language.

**Knowledge graph schema.** We summarize the node and edge types in Tables 7 and 8 and provide complete examples of user profile memory instances in Figures 12 and 13.

Table 7: Edge schema of the user profile memory, including relation types and JSON-style examples.

| Edge Type | Source | Relation | Target | Example |
|---|---|---|---|---|
| Hierarchical | user | refers_to | knowledge | { "source": "u1", "target": "k1", "type": "Hierarchical", "relation": "refers_to" } |
| Hierarchical | knowledge | entails | pattern | { "source": "k1", "target": "p1", "type": "Hierarchical", "relation": "entails" } |
| Hierarchical | pattern | target | object | { "source": "p1", "target": "o1", "type": "Hierarchical", "relation": "target" } |
| Hierarchical | pattern | target | location | { "source": "p1", "target": "l1", "type": "Hierarchical", "relation": "target" } |
| Hierarchical | knowledge | composed_of | object | { "source": "k1", "target": "o2", "type": "Hierarchical", "relation": "composed_of" } |
| Temporal | pattern | before | pattern | { "source": "p1", "target": "p2", "type": "Temporal", "relation": "before" } |

Table 8: Node schema of the user profile memory.

| Node Type | Attributes | Example |
|---|---|---|
| User | id, type, name | {u1, "user", "james"} |
| Knowledge | id, type, subtype, alias, description | {k_obj1, "knowledge", object_semantics, "my favorite cup", "a yellow cup with a wooden handle"} |
| Pattern | id, type, name, args | {p1, "pattern", "place", [cup, on, livingroom_table]} |
| Object | id, type, name, granularity | {o2, "object", "red dog image cup", instance} |
| Location | id, type, name, expression | {l1, "location", livingroom_table, "on the table"} |

### Object Semantic Knowledge: James' dinnerware set

```
{
  "nodes": [
    { "id": "u1", "type": "User", "name": "James"},
    { "id": "k1", "type": "Knowledge", "subtype": "object semantics", "alias": "my dinnerware set"},
    { "id": "o1", "type": "Object", "name": "plate", "granularity": "instance",
      "attributes": ["square", "white", "black geometric pattern"]},
    { "id": "o2", "type": "Object", "name": "bowl", "granularity": "instance",
      "attributes": [ "white", "brown lid"]}
  ],
  "edges": [
    {"source": "u1", "target": "k1", "type": "Hierarchical", "relation": "refers to"},
    { "source": "k1", "target": "o1", "type": "Hierarchical", "relation": "target"},
    { "source": "k1", "target": "o2", "type": "Hierarchical", "relation": "target"}
  ]
}
```

Figure 12: Example of a user profile memory graph representing object semantics knowledge.

**User Pattern Knowledge: James' living room play setup**

```
{
  "nodes": [
    {"id": "u1", "type": "User", "name": "James"},
    {"id": "k1", "type": "Knowledge", "subtype": "user pattern", "alias": "living room play
        setup", "description": "keep toy vehicle on the living room couch so it's easy to find
        when playing"},
    {"id": "p1", "type": "Pattern", "name": "move", "args": ["toy vehicle", "from bedroom", "to
        living room"]},
    {"id": "p2", "type": "Pattern", "name": "place", "args": ["toy vehicle", "on couch", "in living
        room"]},
    {"id": "o1", "type": "Object", "name": "toy vehicle"},
    {"id": "l1", "type": "Location", "name": "bedroom", "expression": "in the bedroom"},
    {"id": "l2", "type": "Location", "name": "livingroom", "expression": "in the living room"},
    {"id": "l3", "type": "Location", "name": "couch", "expression": "on the couch in the living
        room"}
  ],
  "edges": [
    {"source": "u1", "target": "k1", "type": "Hierarchical", "relation": "refers to"},
    {"source": "k1", "target": "p1", "type": "Hierarchical", "relation": "entails"},
    {"source": "k1", "target": "p2", "type": "Hierarchical", "relation": "entails"},
    {"source": "p1", "target": "o1", "type": "Hierarchical", "relation": "target"},
    {"source": "p1", "target": "l1", "type": "Hierarchical", "relation": "target"},
    {"source": "p1", "target": "l2", "type": "Hierarchical", "relation": "target"},
    {"source": "p2", "target": "o1", "type": "Hierarchical", "relation": "target"},
    {"source": "p2", "target": "l3", "type": "Hierarchical", "relation": "target"},
    {"source": "p1", "target": "p2", "type": "Temporal", "relation": "before"}
  ]
}
```

Figure 13: Example of a user profile memory graph representing user pattern knowledge.

## D.2    SUCCESS AND ERROR CASE ANALYSIS

As shown in Figure 5, we sampled success and error cases in memory utilization stage.

**Tasks require object semantics knowledge.**    Success cases demonstrate that agents can effectively reference personalized object attributes from episodic memory, correctly identifying and applying the specific information needed for task completion. (**A.1**) However, we observed distinct error patterns when agents needed to utilize this type of knowledge: the most prominent being commonsense reasoning errors (**B.1**), where agents incorrectly relied on generic object descriptions instead of personalized cues; missed personalization cues (**B.2**), where agents failed to recognize the need to access personalized knowledge; and memory recall failures (**B.3**), where agents were unable to locate relevant information despite its presence in the provided context.

**Tasks require user patterns.**    For user patterns tasks, we observed that agents employed two distinct strategies: commonsense reasoning (**C.1**) treating memory as exemplars for step-by-step reasoning, and direct reference (**C.2**) for distinctive patterns like "my go-to breakfast." However, both approaches introduced specific vulnerabilities leading to two common failure patterns. First, commonsense reasoning (**D.1**) happened when agents attempted to apply the reasoning-based approach but encountered gaps they couldn't bridge, leading them to substitute commonsense knowledge that seemed plausible but contradicted established personalized routines. Second, inaccurate recall (**D.2**) occured when agents recognized the need for personalized knowledge but retrieved imprecise or incomplete information from memory. The sequential nature of these approaches made them particularly error-prone, as mistakes at any intermediate step propagated through subsequent actions. This vulnerability explains the consistently higher failure rates in user patterns tasks across all models, highlighting fundamental challenges in maintaining coherence through multi-step reasoning over episodic memory.

### D.3 MEMORY DESIGN EXPERIMENT DETAILS

We describe here the experimental setup for Section 6. The following details clarify the episode sampling, memory construction, and retrieval configurations used in this experiment.

#### D.3.1 DETAILS OF MEMORY FORMAT SIMPLIFICATION EXPERIMENT

We sampled 30 episodes by selecting 10 episodes from each of three scenes, with careful balancing across task types and difficulty levels informed by our preliminary experimental results. To minimize the influence of erroneous trajectories, we provided GPT-4o-generated gold memory as the given episodic memory, thereby reducing the impact of noisy interaction histories. We also used GPT-4o to generate summaries of the action trajectories, including user instructions, to facilitate compact memory representations. For each evaluation, one exemplar shot was given to the agent, and top-$k$ memory entries were retrieved based on the current query, where $k = 10$ for frontier models (GPT-4o and Qwen-2.5-72B) and $k = 7$ for smaller models (LLaMA-3.1-8B and Qwen-2.5-7B), considering context length limitations. The prompts used for these experiments are provided in Appendix F.3.

#### D.3.2 DETAILS OF USER PROFILE MEMORY EXPERIMENT

We conduct experiments by incorporating user profile memory in addition to episodic memory. The experiments are conducted with the same models as in Section 6.1 (GPT-4o, Qwen-2.5-72B-Instruct, Llama-3.1-8B-Instruct, and Qwen-2.5-7B-Instruct). These models are also employed both to construct the user profile memory and to carry out the retrieving procedure, which combines embedding-based search with LLM reasoning: the LLM parses user instructions and reformulates retrieved graph content into natural language descriptions. User profile memory entries are retrieved under a $k = 5$ setup without gold memory, while episodic memory is supplied as each agent's trajectory under a $k = 5$ setup with gold memory. We evaluate performance on both the single and joint memory utilization stages, and the prompts for memory construction and retrieval are provided in Appendix F.

### D.4 EXAMPLE TRAJECTORY AND EVALUATION

To provide readers with a concrete understanding of our evaluation framework and typical agent performance patterns, we present detailed examples of both successful (Tables 9, 10, 11) and failure cases (Tables 12, 13, 14). These examples include the complete instruction metadata, agent execution trajectories, and evaluation outcomes, offering insights into when and why agents succeed or struggle with personalized knowledge utilization.

Table 9: Episode metadata for the example task.

| Field | Value |
|---|---|
| episode_type | user_pattern |
| subtype | preference |
| task | Could you decorate the bedroom as I like? |
| task_percent_complete | 1.0 |
| task_state_success | 1.0 |

Table 10: Trajectory for the example task.

| # | Thought | Action | Result / State $\Delta$ |
|---|---------|--------|-------------------------|
| 1 | Since there are no objects found yet, I should start by exploring the bedroom to locate any objects that can be used for decoration. | Explore (bedroom_1) | No objects found |
| 2 | Since no objects were found in bedroom_1, I should explore another room to locate objects that can be used for decoration. I will explore the living room next, as it is likely to contain decorative items. | Navigate (to living_room_1) | Location changed |
| 3 | I have found a plant container and a statue on the shelves in the living room, which can be used for decoration in the bedroom. I will start by picking up the plant container from the shelves. | Pick (plant_container_1) | plant_container_1 held |
| 4 | Now that I am close to the shelves, I will attempt to pick up the plant container to move it to the bedroom for decoration. | (continue) | Success |
| 5 | I have successfully picked up the plant container. Now, I need to navigate to the bedroom to place it for decoration. | Navigate (to bedroom_1) | Arrived in bedroom_1 |
| 6 | I have successfully navigated to bedroom_1. Now, I need to find a suitable location to place the plant container for decoration. I will place it on the chest of drawers in the bedroom. | Place (plant_container_1 on chest_of_drawers_54) | Object placed |
| 7 | Now that I am close to chest_of_drawers_54, I will place the plant container on it for decoration. | (continue) | Confirmed placement |
| 8 | I have successfully placed the plant container on chest_of_drawers_54 in bedroom_1. Now, I need to return to the living room to pick up the statue from the shelves for further decoration in the bedroom. | Navigate (to living_room_1) | Location changed |
| 9 | I am back in the living room. I need to navigate closer to the shelves to pick up the statue for further decoration in the bedroom. | — | Positioning for pick |
| 10 | Now that I am close to the shelves, I will attempt to pick up the statue to move it to the bedroom for decoration. | Pick (statue_0) | statue_0 held |
| 11 | I have successfully picked up the statue. Now, I need to navigate to the bedroom to place it for decoration. | Navigate (to bedroom_1) | Location changed |
| 12 | I have successfully navigated to bedroom_1. Now, I need to find a suitable location to place the statue for decoration. I will place it on the chest of drawers next to the plant container. | — | Positioning |
| 13 | Now that I am close to chest_of_drawers_54, I will place the statue next to the plant container for decoration. | Place (statue_0 on chest_of_drawers_54 next_to plant_container_1) | Object placed |

Table 11: Evaluation functions for the example task.

| # | Function | Args | Description |
|---|---|---|---|
| 1 | is_on_top | object_handles=[Cole_Hardware_Electric_Pot_Assortment_55_:0000], receptacle_handles=[bedroom_chest_of_drawers_handles], number=1 | Plant container is on drawer |
| 2 | is_on_top | object_handles=[Dog_Statue_4_:0000], receptacle_handles=[bedroom_chest_of_drawers_handles], number=1 | Statue is on drawer |
| 3 | is_next_to | entity_handles_a=[Cole_Hardware_Electric_Pot_Assortment_55_:0000], entity_handles_b=[Dog_Statue_4_:0000], number=1, l2_threshold=1.0 | Plant and statue are next to each other |

Table 12: Episode metadata for the example task.

| Field | Value |
|---|---|
| episode_type | user_pattern |
| subtype | routine |
| task | Could you set up the candle for a cozy evening? |
| task_percent_complete | 0 |
| task_state_success | 0 |

Table 13: Trajectory for the example task.

| # | Thought | Action | Result / State $\Delta$ |
|---|---|---|---|
| 1 | Since there are no objects found yet, I should start by exploring the tv room to locate a candle that can be set up for a cozy evening. | Navigate (to tv_1) | Location changed |
| 2 | I have found a candle, candle_0, on table_34 in the tv room. To set it up for a cozy evening, I will first need to pick it up from the table. | Pick (candle_0) | candle_0 held |
| 3 | Now that I am close to table_34, I will attempt to pick up the candle, candle_0, to set it up for a cozy evening. | (continue) | Success |
| 4 | I have successfully picked up the candle, candle_0. To set it up for a cozy evening, I will place it on a suitable surface in the tv room, such as the tv table, table_14. | Place (candle_0 on table_14) | Object placed |
| 5 | I am now close to table_14 in the tv room. I will place the candle, candle_0, on this table to set it up for a cozy evening. | (continue) | Failed to meet task constraints |

Table 14: Evaluation functions for the example task.

| # | Function | Args | Description |
|---|----------|------|-------------|
| 1 | is_on_top | object_handles=[candle], receptacle_handles=[chair_handles], number=1 | Candle must be on chair |
| 2 | is_on_top | object_handles=[candle], receptacle_handles=[table_handles], number=1 | Candle must be on table (after chair) |
| 3 | Constraint | TemporalConstraint: Placing the candle on the chair before placing it on the table. | |

### D.5 LLMs LONG CONTEXT COMPREHENSION ANALYSIS

To investigate whether the *information overload* bottleneck in LLM-powered embodied agents for personalized assistance stems from limitations in processing long-context information, we conduct an experiment in a QA format to evaluate whether LLMs can properly reference personalized information from long contexts.

**Experiment setup.** We provide LLMs with the episodic memory information required for personalized assistance tasks along with the corresponding task instructions. We then prompted the models: "Identify the personalized knowledge necessary to perform this instruction based on the context provided below." To evaluate the responses, we use GPT-4o as an LLM-as-a-Judge, supplying it with the stored gold memory and task instruction as references to assess whether the LLM's identified personalized knowledge aligns with that in the gold memory.

Table 15: Accuracy on long-context QA with varying numbers of retrieved episodic memories (Top-$k$).

| Model | k = 1 | 3 | 5 | 7 | 10 | 20 |
|-------|-------|---|---|---|-----|-----|
| GPT-4o | 100% | 98.71% | 98.28% | 96.98% | 96.98% | 95.69% |
| Qwen-2.5-72b | 100% | 93.97% | 94.40% | 93.53% | 93.10% | 83.19% |

**Results.** As shown in the Table 15, all models demonstrate high accuracy in correctly identifying personalized knowledge in the QA format. Compared to the results in Section 5.1 Figure 6, this indicates that current LLMs have the ability to reference personalized information from long contexts, but encounter difficulties when applying this personalized information to planning tasks.

## E EXTENDED ANALYSIS: MEMORY COMPONENTS AND REAL-WORLD SCENARIOS

To provide comprehensive insights into memory utilization for personalized assistance, we conduct extended analysis examining multiple factors affecting agent performance. This section covers three primary areas: memory quality effects on agent performance, retrieval strategy effectiveness for episodic memory, and agent behavior in complex real-world scenarios including 3-joint memory tasks and ambiguous instructions.

### E.1 HUMAN BASELINE

**Experiment setup.** To establish a meaningful comparison point for our agent evaluation and validate that our tasks are fundamentally solvable, we conduct a human baseline study. We recruit four undergraduate participants to complete a curated set of 20 tasks that represent the challenges our embodied agents face in memory utilization for personalized assistance.

To ensure comprehensive coverage of task complexity, we categorize task difficulty based on empirical observations of model behavior across our agent evaluations:

- **Easy**: Llama-8B successfully completed the task using its own memory

- **Medium**: Llama-8B failed with its own memory but succeeded when given GPT-4o's answer memory

- **Hard**: Llama-8B failed even when given GPT-4o's answer memory

- **Super hard**: GPT-4o failed with its own memory

- **Random joint-memory tasks**

We implement a text-based interactive interface that mirrors the agent environment while accommodating human interaction patterns. Participants issue step-by-step commands and receive observational feedback after each action, similar to how our agents operate. Each participant receives a 10-minute tutorial on the interface and completes tasks in the same environment as the agents. Crucially, participants have access to the same top-5 retrieved memories that agents receive, allowing direct comparison of memory utilization capabilities.

**Results.** The human baseline provides both validation and perspective on our evaluation framework. All participants successfully completed 100% of the tasks with correct end states, demonstrating that the challenges are fundamentally solvable given appropriate memory access and reasoning capabilities. While humans require more time than LLM-based planners, they exhibit superior abilities in referencing episodic memory, recognizing errors, and adapting their actions accordingly. This performance establishes a meaningful upper-bound reference for agent capabilities and confirms that our tasks capture genuine challenges in memory utilization for personalized assistance.

## E.2 MEMORY QUALITY AND ROBUSTNESS ANALYSIS

To better understand the factors affecting memory utilization performance, we conduct additional analysis examining two critical aspects of memory-based personalized assistance. First, we analyze how memory quality impacts agent performance by comparing high quality memory trajectories with acquired memory trajectories from actual agent interactions. Second, we evaluate agent robustness when episodic memories are corrupted or incomplete through various noise and degradation conditions. These analyses provide insights into the practical limitations and requirements for deploying personalized embodied agents in real-world scenarios where memory quality cannot be guaranteed.

### E.2.1 MEMORY QUALITY ANALYSIS

**Experiment setup.** We further analyze the effect of memory quality by comparing high quality memory, consisting of successful and shortest-path trajectories that serves as curated references with the memory obtained from interaction histories in the *memory acquisition stage*, with predicted memory obtained via automated retrieval. By evaluating performance differences between these two settings, we aim to understand how memory quality affects agent performance across the *memory utilization stage* for both tasks (single-memory tasks, joint-memory tasks).

**Performance degradation with lower-quality trajectories.** Table 16 shows the performance comparison between high quality memory and acquired memory across the *memory utilization stage*. In the *memory utilization stage*, high-capacity models show relatively stable performance across gold and retrieved memory, while lower-capacity models exhibit a substantial drop when using retrieved memory. This suggests that less capable models have more difficulty extracting relevant information from an imperfect memory context. When executing more demanding joint-memory tasks, which require combining and reasoning across multiple memory sources, performance degrades sharply across all models with acquired memory. This indicates that the compounded complexity of integrating multiple memories makes it increasingly important for the agent to rely on clear, noise-free, high-quality memory contexts from which synthesis can be performed reliably. Additionally, as shown in the table X, the overall reduction in planning steps when using high-quality memory suggests that episodic memory plays an important role in the planning process of embodied agents. Enhancing memory quality through more precise filtering is therefore essential not only for improving reasoning performance but also for increasing the agent's planning efficiency.

Table 16: Performance comparison of models when using high-quality memory versus acquired (self-acquired) memory, evaluated in both single- and joint-memory settings.

| Model | Memory | Stage | Replanning Count ↓ | Sim Steps ↓ | PC (%) ↑ | SR (%) ↑ |
|---|---|---|---|---|---|---|
| GPT-4o | High-quality | Single Memory | 14.5 | 2481.2 | 85.3 | 80.6 |
| | | Joint Memory | 25.8 | 3619.4 | 83.3 | 63.9 |
| | Baseline | Single Memory | 16.1 | 2450.8 | 87.9 | 85.1 |
| | | Joint Memory | 28.9 | 3480.7 | 86.7 | 63.9 |
| Claude-3.5-Sonnet | High-quality | Single Memory | 14.1 | 2266.4 | 77.2 | 70.1 |
| | | Joint Memory | 24.2 | 3226.6 | 69.9 | 38.9 |
| | Baseline | Single Memory | 15.3 | 2258.8 | 69.3 | 63.7 |
| | | Joint Memory | 27.8 | 3198.8 | 64.6 | 33.3 |
| Qwen-2.5-72B | High-quality | Single Memory | 15.2 | 2630.4 | 74.7 | 68.2 |
| | | Joint Memory | 26.4 | 4082.1 | 64.8 | 38.9 |
| | Baseline | Single Memory | 17.5 | 2691.2 | 72.6 | 67.2 |
| | | Joint Memory | 31.3 | 4027.1 | 68.9 | 36.1 |
| Llama-70B | High-quality | Single Memory | 15.1 | 2364.0 | 72.6 | 67.2 |
| | | Joint Memory | 27.1 | 3374.5 | 64.1 | 27.8 |
| | Baseline | Single Memory | 19.0 | 2566.6 | 72.2 | 66.7 |
| | | Joint Memory | 31.4 | 3425.2 | 51.3 | 8.3 |
| Llama-8B | High-quality | Single Memory | 16.9 | 2739.0 | 65.1 | 59.2 |
| | | Joint Memory | 26.4 | 3753.2 | 54.0 | 13.9 |
| | Baseline | Single Memory | 19.0 | 3131.7 | 48.1 | 35.0 |
| | | Joint Memory | 27.4 | 3478.2 | 35.3 | 8.3 |
| Qwen-2.5-7B | High-quality | Single Memory | 15.7 | 2991.5 | 55.7 | 46.3 |
| | | Joint Memory | 23.9 | 4402.0 | 34.0 | 8.3 |
| | Baseline | Single Memory | 21.8 | 3271.4 | 39.1 | 27.4 |
| | | Joint Memory | 26.9 | 4148.6 | 33.7 | 5.6 |

Table 17: Model performance across memory acquisition and utilization stages in MEMENTO.

| Model | Memory | Memory Type | Planning Cycles ↓ | Sim Steps ↓ | Percent Complete ↑ | ΔPC | Success Rate ↑ | ΔSR |
|---|---|---|---|---|---|---|---|---|
| GPT-4o | Gold | Baseline | 14.5 | 2481.2 | 85.3 | – | 80.6 | – |
| | Noisy | Random | 14.6 | 2759.0 | 81.5 | **-3.8** | 78.6 | **-2.0** |
| | | Shuffle | 14.4 | 2450.9 | 84.0 | -1.3 | 80.6 | 0.0 |
| | Partial | Random | 14.5 | 2526.9 | 84.4 | -0.9 | 80.1 | -0.5 |
| | | w/o Pick | 14.6 | 2555.7 | 82.9 | -2.4 | 79.1 | -1.5 |
| | | w/o Place | 14.5 | 2569.8 | 84.4 | -0.9 | 82.6 | +2.0 |
| | | w/o Nav | 15.4 | 2526.8 | 81.4 | **-3.9** | 78.5 | **-2.1** |

### E.2.2 ROBUSTNESS TO NOISY AND PARTIAL MEMORIES

**Experiment setup.** We conduct additional experiments to evaluate the model's robustness when episodic memories are corrupted or incomplete. As the target model, We leverage GPT-4o, and we organize the experimental setup into two representative degradation types: *Noisy Memory* and *Partial Memory*. For *Noisy Memory*, we consider two conditions: (i) random step injection, where unrelated steps from other scenes (30%) are inserted into the memory trajectory to simulate retrieval or perception noise, and (ii) temporal shuffle, where the step order within an episode is randomly permuted to disrupt temporal coherence. For *Partial Memory*, we examine two forms of incompleteness: (i) random step removal, where 20% of steps were randomly removed to test agent robustness under incomplete recall, and (ii) action-type-specific removal, where all steps of a given action type (Place, Place, Navigate) are dropped to analyze their impact on planning.

**Random noise and missing navigation cues degrade performance.** As shown in Table 17, the model remains robust under most forms of incomplete memory, with only minor degradation. However, two factors consistently cause substantial performance drops: (i) the injection of random,

noisy entries and (ii) the absence of navigation-related information. This indicates that the model is more sensitive to misleading or corrupted inputs than to simple omissions, and that navigation cues play a disproportionately critical role in planning.

### E.3    RETRIEVAL STRATEGY ANALYSIS

Effective retrieval of relevant episodic memories is fundamental to successful memory utilization in personalized assistance tasks. To understand how retrieval quality affects agent performance, we analyze both baseline retrieval effectiveness and the impact of advanced retrieval strategies on downstream task outcomes. We first establish baseline missing rates using similarity-based retrieval to determine optimal retrieval parameters, then examine whether advanced retrieval strategies consistently improve both memory retrieval quality and agent task performance. This analysis provides insights into the relationship between retrieval precision and practical task execution in personalized embodied agents.

#### E.3.1    BASELINE RETRIEVAL ANALYSIS.

To assess whether the corresponding episodic memory is successfully retrieved during memory utilization, we first measure the missing rate under a baseline similarity-based retriever. [1] The results show missing rates of 13.9% for top-3 and 3.5% for top-5, indicating that expanding to top-5 nearly eliminates missing cases. Based on this observation, we set top-5 as the default retrieval setup, prioritizing inclusion of the required memory over the risk of distractor entries.

Table 18: We use an embedding-based retrieval method with the *all-mpnet-base-v2* Sentence Transformer (Reimers & Gurevych, 2019).

| Retriever Strategy | Top-K | Missing Rates (%) |
|---|---|---|
| Baseline | 3 | 13.9 |
|  | 5 | 3.5 |

#### E.3.2    ADVANCED RETRIEVAL STRATEGY ANALYSIS

Table 19: Performance comparison across different retrieval strategies.

| Model | Strategy | Top-K | Missing Rates (%) | Percent Complete ↑ | Success Rate ↑ |
|---|---|---|---|---|---|
| GPT-4o | Baseline | 3 | 13.9 | 82.9 | 79.5 |
|  |  | 5 | 3.5 | 84.7 | 82.1 |
|  | Reranker | 3 | 2.5 | 87.2 | 84.1 |
|  |  | 5 | 1.5 | 84.3 | 80.6 |
|  | Query Expansion | 3 | 9.0 | 84.9 | 81.1 |
|  |  | 5 | 1.5 | 86.2 | 81.1 |
| Qwen-2.5-72b | Baseline | 3 | 13.9 | 75.3 | 70.1 |
|  |  | 5 | 3.5 | 75.4 | 70.0 |
|  | Reranker | 3 | 7.6 | 73.0 | 67.2 |
|  |  | 5 | 2.1 | 70.9 | 64.1 |
|  | Query Expansion | 3 | 8.5 | 75.6 | 69.2 |
|  |  | 5 | 4.0 | 72.6 | 66.2 |

**Experiment setup.**    Building on this baseline, we further examine the effect of advanced retrieval strategies on the instruction following capability. We compared three approaches: (i) the baseline similarity-based retrieval, (ii) reranker-based retrieval, which applies an LLM-based pointwise rerranker (Sachan et al., 2022; Sun et al., 2023) to an initial pool of 10 candidates before selecting the final top-3 or top-5, and (iii) query expansion-based retrieval, where queries are reformulated before retrieval (Jagerman et al., 2023; Wang et al., 2023b). These strategies are evaluated with both GPT-4o and Qwen-2.5-72b, using the same models to compute reranker scores and produce expanded queries.

---

[1]We adopt an embedding-based method using the *all-mpnet-base-v2* Sentence Transformer (Reimers & Gurevych, 2019).

**Improved retrieval does not ensure improved outcomes.** As shown in Table 19, retrieval strategies affect not only memory inclusion but also downstream task performance. First, applying reranker or query expansion substantially reduce the missing rate, particularly under the top-3 setting, where improvements range from 4.9pp to 11.4pp. This confirms their effectiveness in narrowing the gap between top-3 and top-5 retrieval. Second, performance gains arere not consistent across models. For example, with Qwen, query expansion under the top-5 setting even degraded retrieval, suggesting that the effectiveness of query expansion critically depends on the quality of the model generating the reformulated queries. Finally, except for GPT-4o at top-3, introducing advanced retrieval strategies further reduces downstream performance, likely due to the inclusion of semantically similar but task-irrelevant memories. These results suggest that simply increasing retrieval relevance does not necessarily yield better task performance, particularly in real-world contexts with dense and noisy memory pools.

### E.4 REAL-WORLD SCENARIO ANALYSIS

While our main evaluation demonstrates fundamental limitations in memory utilization for personalized assistance, real-world deployment introduces additional complexities that extend beyond our controlled experimental settings. To explore these practical challenges, we conduct three sets of experiments that examine agent behavior under more realistic conditions. First, we evaluate scalability by testing agents on 3-joint memory tasks that require integrating knowledge from multiple episodes. Second, we assess how agents handle ambiguous instructions that indirectly reference personalized knowledge through contextual cues rather than explicit mentions. Finally, we investigate agents' ability to generalize learned user preferences across different but related contexts, such as applying spatial arrangements from one room to another. These experiments provide insights into additional hurdles for successful real-world deployment of personalized embodied agents.

#### E.4.1 REAL-WORLD MULTI-EPISODE EVALUATION

In practice, user requirements are rarely limited to information from just one or two episodes, but often require knowledge distributed across multiple episodes. To explore this realistic scenario, we extend our joint-memory task to cases involving three memories, which require agents to effectively integrate three independent memory sources to solve the task. Following the same methodology

Table 20: Performance results on multiple joint-memory tasks.

| Model | Task | PC (%) | SR (%) |
|---|---|---|---|
| GPT-4o | joint memory task | 86.7 | 63.9 |
| GPT-4o | 3-joint memory task | 61.2 | 22.0 |
| Qwen-2.5-72b | joint memory task | 68.9 | 36.1 |
| Qwen-2.5-72b | 3-joint memory task | 42.4 | 7.3 |

as our joint-memory task construction, we sequentially concatenated three personalized instructions to generate a total of 41 tasks. We observed substantial performance drops compared to the two-memory joint-memory tasks for both GPT-4o and Qwen-2.5-72b-instruct. This suggests that multi-memory reasoning introduces significant difficulty due to temporal gaps and abstraction challenges. We believe this degradation reflects the true complexity of real-world personalized tasks, and addressing this challenge remains critical for future research.

#### E.4.2 AGENT BEHAVIOR UNDER AMBIGUOUS INSTRUCTIONS FROM USERS

**Experiment setup.** While MEMENTO focuses on evaluating personalized knowledge grounding with explicit references to prior interactions, real-world human-agent communication often involves ambiguous or indirect references. To explore this challenge, we conducted a proof-of-concept experiment to assess whether current models can interpret ambiguous instructions that indirectly refer to previously encountered personalized knowledge. We created a tailored set of 30 tasks by sampling 10 episodes from each of three scenes and modifying them to include indirect references

Table 21: Performance under ambiguous queries for personalized knowledge. PC (Percent Complete) and SR (Success Rate) (%) indicate how well agents resolve indirect references to personalized knowledge from memory.

| Model | PC (%) | SR (%) |
|---|---|---|
| GPT-4o (Baseline) | 92.0 | 90.0 |
| GPT-4o | 80.4 | 73.3 |
| Qwen-2.5-72b (Baseline) | 75.1 | 66.7 |
| Qwen-2.5-72b | 59.6 | 53.3 |

to personalized knowledge through contextual cues, synonyms, or causal references (*e.g.*, *Can you set my afternoon tea time routine?* → *I'm about to enjoy my afternoon tea. Could you set things*

*up as I like them?*). To ensure quality and mitigate potential bias, all episodes were jointly created and validated by two authors. For each model, we first generated memory traces during the memory acquisition stage and then evaluated memory usage during the utilization stage. Table 22 provides representative examples of the ambiguous instruction pairs used. In each case, the user rephrases the original request to refer indirectly to previously established personalized knowledge, simulating natural variability in real-world human-agent communication.

**Result.** The results, shown in Table 21, reveal the degradation in performance, indicating that handling ambiguous queries remains a key challenge for future personalized embodied agents. We view this as a promising direction for future work, where we plan to systematically extend MEMENTO to evaluate LLM-powered embodied agents' capabilities under ambiguous and implicit reference scenarios, aiming to better reflect the complexity of real-world human-agent interaction.

Table 22: Examples of ambiguous instruction modifications in the additional experiment dataset.

| Original Memory Utilization Instruction | Rephrased Ambiguous Instruction |
|---|---|
| Can you arrange **my cozy reading setup on the dining table**? | Can you **set up the dining table as I prefer** so I can **read the book comfortably**? |
| Could you help me tidy up by moving **my graduation gifted book**, the candle with a brown and blue gradient, and the round white clock with black numbers to the stand? | Could you help me tidy up by moving **my book received to celebrate completing my studies**, the candle with a brown and blue gradient, and the round white clock with black numbers to the stand? |

### E.4.3 PERSONALIZED KNOWLEDGE GENERALIZATION

**Experiment setup.** A critical aspect of personalized embodied agent is their ability to generalize learned user preferences across different but related contexts. This includes applying knowledge about a user's breakfast routine to dinner preparation tasks, or transferring spatial preferences from one room to another. To investigate this capability, we designed generalization tasks that test whether embodied agents can abstract and reuse personalized knowledge as transferable templates. We extended our user pattern tasks by sampling existing scenarios and systematically altering target locations, thereby evaluating whether agents can adapt user preferences to novel but structurally similar situations. To encourage explicit memory utilization, we incorporated contextual cues in the instructions, such as "on [another receptacle] this time," clearly signaling that established preferences should apply in the modified context. Our generalization tasks were constructed using relatively simple base cases that GPT-4o consistently solves in standard settings.

**Result.** As in Table 23, we observed that GPT-4o occasionally fails by defaulting to commonsense reasoning rather than retrieving and adapting the relevant personalized memory. Interestingly, Qwen-2.5-72b demonstrated improved completion rates on these tasks. We attribute this improvement to the explicit instruction cues, which likely reduced ambiguity and guided the model

Table 23: Agent performance on personalized knowledge generalization tasks.

| Model | Task | PC (%) | SR (%) |
|---|---|---|---|
| GPT-4o | Baseline | 100.0 | 100.0 |
| GPT-4o | Generalization | 92.5 | 83.3 |
| Qwen-2.5-72b | Baseline | 75.0 | 75.0 |
| Qwen-2.5-72b | Generalization | 85.8 | 75.0 |

toward memory-based reasoning. While these preliminary experiments provide initial insights into generalization capabilities, they represent an important step toward understanding how agents can flexibly apply personalized knowledge. Future work could build on these initial observations to better understand how agents transfer personalized knowledge across different contexts.

### E.4.4 USER PROFILE MEMORY ROBUSTNESS

**Experiment setup.** In real-world interactions, users often refer to knowledge, objects, or locations using different names or expressions rather than clean and consistent symbolic identifiers. To directly evaluate the robustness of our proposed User Profile Memory under realistic conditions, we additionally conduct a qualitative analysis on how well User Profile Memory handles natural noise that arises in real user interactions. We identified two noise types that naturally arise when users interact with a personalized embodied agent:

- Knowledge Update Noise - the user expresses an existing knowledge using a different phrase (synonyms, paraphrases, alias names).
- Node Reference Noise - new knowledge refers to an existing object / location using a different form.

For each category, three undergraduate annotators modified the original instructions grounded in the scene of MEMENTO, producing 10 noisy instructions for each noise type. This procedure ensures the noise realistically reflects how humans describe object semantics or user patterns. These noisy instructions were crafted using the underlying scene (103997895_171031182), and to ensure an accurate evaluation of noise robustness, all experiments were conducted on graphs constructed from the original instructions. Full datasets are provided in Table 24 and Table 25

Table 24: Dataset of Knowledge Update Noise Variants.

| Original Knowledge | Modified Knowledge |
|---|---|
| Gift teapot from grandfather | Grandfather's gifted teakettle |
| Family heirloom vase | Inherited family vase |
| Gift toy airplane from childhood friend | A toy airplane that a childhood friend gave me |
| Favorite relaxation candle | The candle I always use to relax |
| Collectible android figure | Android collectible figurine |
| Bedroom organization | Organizing setup for the bedroom space |
| Morning work session setup | Morning workspace arrangement |
| Cozy room setup | Arrangement for creating a cozy room atmosphere |
| Bedtime setup | Nighttime setup before going to bed |
| Study materials organization | Organized placement of study materials |

Table 25: Dataset of Node Reference Noise Variants.

| Original Description | Modified Description |
|---|---|
| Black teapot with curved spout and ornate lid | A curved-spout black teapot featuring an intricately decorated lid |
| Gray laptop with black keyboard/touchpad | A gray laptop equipped with a black keyboard and touchpad |
| Dark green analog alarm clock with twin bells | Dark green twin-bell analog alarm clock |
| Black and orange keychain with circular pendant | A circular pendant keychain in black and orange |
| android figure which is a black and white android panda figure | a black and white Android panda figurine |
| Move the toy airplane from the bedroom to the living room table. Place it next to the couch. | Relocate the toy plane from the bedroom onto the table in the living room, positioning it next to the sofa. |
| Move the picture frame and clock to the table in the bedroom. Then, place them next to each other on the counter in the kitchen. | Bring the photo frame and the alarm clock to the bedroom table, and afterward arrange both items side by side on the countertop at kitchen. |
| Place the bread, the potato, and the egg on the kitchen table and place them next to each other. | Set the loaf, spud, and fresh egg together on the table at kitchen, arranging them close to one another. |
| Move the laptop and the mouse pad to a new living room table. Move the clock to another shelf. | Transfer the notebook computer and mouse mat onto the new table in the living room, and place the clock on a separate shelf. |
| Take the book, pencil case, and multiport hub from the table in the living room and put them on the shelves in the bedroom, placing them next to each other. | Gather the note, pencil pouch, and usb hub from the living room table and arrange them side by side on the bedroom shelves. |

**Result.** As shown in Table 26, knowledge update/add decisions are handled reliably across all samples. These findings indicate that our method handles natural linguistic noise reasonably well. However, node reference shows a few missed cases (5 out of 20). Although these missed cases amount to 25%, the model still correctly handles 75% of the reference variations. The specific failure cases are summarized in Table 27. These failures indicate that reducing unnecessary node duplication remains an important challenge that requires further refinement. We believe this direction is important because mitigating node duplication directly improves memory consistency, which is a key factor influencing downstream task performance in personalized embodied agents.

Table 26: Accuracy under different noise types.

| Noise Type | Accuracy (%) |
|---|---|
| Knowledge Update Noise | 100.0 |
| Node Reference Noise | 75.0 |

Table 27: Failure cases in node reference noise

| Existing Node | Add Node |
|---|---|
| mouse pad | mouse mat |
| clock | alarm clock |
| android figure which is a black and white android panda figure | black and white Android Panda Figurine |
| kitchen counter | countertop at kitchen |
| another shelf | separate shelf |

# F PROMPTS

## F.1 PROMPTS FOR DATASET GENERATION

---

**Object Semantics Instruction Generation Prompt**

object_semantics: |-
  Your task is to generate a user instruction that includes object semantics for an embodied agent that can perform rearrangement tasks.
  The instruction should be grounded in personalized object-level semantics based on the original instruction and object descriptions.

  The object semantics can be categorized into 4 types:
   - ownership: Indicates that the user personally owns or has a special claim on an object.
   - preference: Indicates the user's specific preferences related to an object (e.g., placement, condition).
   - history: Reflects the user's past interaction or meaningful history with the object.
   - group: Defines a logical or personalized grouping of multiple objects (e.g., "my coffee set" for mug + saucer + spoon).

  You should generate 2 types of instructions and object semantics:
   - Stage 1: Instruction for memorization; The instruction should include the original instruction, descriptions of all the objects, and explicit object semantics of the relevant objects. This will be used to store memory.
   - Stage 2: Instruction for utilization; The instruction should require the agent to understand and use the previously stored object semantics. It should sound natural to humans and be difficult for an agent without access to memory. Keep it short and situated. Relevant objects must be referred to only using their stored semantics, without any descriptive attributes. For all other objects, refer to them using visual or descriptive attributes.
   - Object Semantics: This is the semantic information associated with each object used in the instruction. Only include the most relevant one object semantics based on the instruction context. Not all target objects need to be included, but use as many of them as possible.

  Note that if the original instruction involves a sequence of object interactions, that order should be preserved in the Stage 2 instruction.

  The output format should be as follows:

---

[Example]
### Input
- original_instruction: <original instruction>
- handle_info: <list of the objects with short descriptions>

### Output
- Stage 1: <instruction> + <object descriptions> + <object semantics>
- Stage 2: <instruction with object semantics formed in a natural way>
- Used Object: <List about the used objects' categories>
- Object Semantics: <Object semantics category about the relevant objects>

[Example 1]
{shot_examples}
...

### Input
- original_instruction: {instruction}
- handle_info: {handle_info}

---

**User Pattern Instruction Generation Prompt**

user_pattern: |-
  Your task is to generate a user instruction that includes user pattern for embodied agent that can perform rearrangment tasks.
  The instruction should be related to personalized knowledge based on the original instruction.

  The user pattern can be categorized into 2 types:
  - preference: A specific way the user prefers to prepare or arrange their environment when a particular situation occurs.
  - routine: A sequence or setup the user follows as a habit or regular activity.

  You should generate 2 types of instructions, memory, and user pattern:
  - Stage 1: Instruction for memorization; The instruction should be original instruction + user pattern. You should explicitly state the user's preference or routine in the instruction.
  - Stage 2: Instruction for utilization; The instruction should be only about user's preference or routine that a human would naturally use in the situated environment. You should make the instruction difficult for the agent without using memory and try to make it short.

- User pattern: The user pattern should be the user's preference or routine that can be reused for future rearrangement tasks.

Note that if the original instruction requires a sequence of actions, the order of the actions should be followed for the stage 2 instruction.

The output format should be as follows:
### Input
<original instruction>

### Output
- Stage 1: <original instruction> + <user pattern>
- Stage 2: <user pattern formed in a natural way>
- Memory: <Memory about user's preference or routine>
- User pattern: <user pattern>

[Example 1]
{shot_examples}
...

### Input
{org_instruction}

**Captioning Prompt for OVMM Objects**

captioning: >
Generate a short, but precise caption for the given object. Focus only on the object, ignoring the background. Include its type, primary colors, and any distinctive features.

Examples:
{shot_examples}

Image:
Category:
{category}

Image:
{image}

**Captioning Prompt for Captioning_Google Objects**

captioning_google: >
  Generate a short, but precise caption for the given object. Focus only on the object, ignoring the background. Include its type, primary colors, and any distinctive features.
  If you can't recognize the object, refer to the name of the objects I gave.

  Examples:
  {shot_examples}

  Image:
    Category:
    {category}

    Name:
    {name}

    Image:
    {image}

## F.2    PROMPTS FOR AGENT

---

### Zero Shot Agent ReAct Prompt

prompt: |-
     {system_tag}You are an agent that solves embodied-agent planning problems. The task assigned to you will be situated in a house and will generally involve navigating to objects, picking and placing them on different receptacles to achieve rearrangement. You strictly follow any format specifications and pay attention to the previous actions taken in order to avoid repeating mistakes.

     If there are multiple tasks to complete, please follow them in the order they appear in the instruction.

     Rooms do not need to be explored more than once. This means if you have explored the living room and have not found the object, then you should explore the kitchen, if a relevant object is still not found, you should explore the hallway etc...

     Many calls to the same action in a row are a sign that something has gone wrong and you should try a different action.{eot_tag}
{rag_examples}
     {user_tag}Task: {input}

     {world_description}

     Possible Actions:
     {tool_descriptions}
     - Done: Used to indicate that the agent has finished the task. Example (Done[])

     What is the next action to make progress towards completing the task?
     Return your response in the following format

     Thought: <reasoning for why you are taking the next action>
     <next action call>
     Assigned!

---

Here is an example:

Thought: Since there are no objects found I should explore a room I have not explored yet.

Explore[<room name>]

Assigned!

{eot_tag}{assistant_tag}

## F.3 Prompts for Memory Design Experiment

**Summary for Section 6.1 Prompt**

summary: |-
  You are a helpful assistant designed to summarize episodic task
execution traces of an embodied agent.

  You will be given a full trace of the agent's actions, thoughts, and results
as it attempts to follow a human instruction.

  Please output a compact memory paragraph including:
    - Instruction: Copy exactly the instruction from the trace. This is the
sentence just before the first Thought appears. (Try to understand user's
intention well.)
    - Plan: Briefly summarize the key high-level steps the agent performed.

  Guidelines:
    - Use 2 to 3 short sentences.
    - Do not list low-level micro actions.
    - Ignore repeated failures unless they affected the outcome.
    - Do not invent any details not present in the trace.
    - Use past tense and third-person style.

  [Example 1]
   Input:
     {input_trace_example}
   Output:
     {output_example}
   ....

  [Example]
   Input:
     {input_trace}
   Output:

**Summary for Graph Reformulation**

extract_knowledge: |-
 Analyze the following graph and interpret as natural language.

 FOCUS on the user's personalized knowledge and what it is and how we can use it to understand the

 Provide only a simple text description without any formatting or JSON.
 Also provide only the output with any additional explanation.
 For example:

 [Example 1]
 Graph:
  {{
    "knowledge_type": {knowledge_type_example},
    "nodes": {nodes_example},
    "edges": {edges_example}
  }}
 Output: {output_description_example}
 ...

 Graph: {graph}
 Output:

**Summary for Knowledge Classifier**

knowledge_classifier: |-
  You are given an instruction.
  Classify the type of personalized knowledge in the instruction.
  Return ONLY the knowledge type, nothing else.

  The knowledge type should be either "object_semantics" or
"user_pattern".
  object_semantics: Personalized knowledge that the user assigns to
specific objects or groups of objects.
  - Examples: gift, heirloom, favorite item, part of a set, belongs to
someone, has special meaning

  user_pattern: Personalized knowledge that the user assigns to pattern
sequences or behavioral routines.
  - Examples: prefer to keep, often do, like to, setup for later, routine,
habit, pattern of behavior

  Rules:
  - If the instruction only assigns personal meaning to an object →
"object_semantics"
  - If the instruction describes a habit, preference, or routine →
"user_pattern"

  [Example 1]
  Instruction: {instruction_example}
  Output: {output_knowledge_type_example}
  ...

  Instruction: {instruction}
  Output:

**Summary for Graph Encoder**

graph_encoder: |-
  You are given an instruction and its knowledge type.
  Generate a semantic graph JSON representing the personalized knowledge.
  Do not output plain text or additional explanations.

  Knowledge Type: {knowledge_type}
  object_semantics: Personalized knowledge that the user assigns to specific objects or groups of objects.
  user_pattern: Personalized knowledge that the user assigns to pattern sequences or behavioral routines.

  For object_semantics: Create Knowledge node + Object nodes (NO Pattern nodes)
  For user_pattern: Create Knowledge node + Pattern nodes + Object/Location nodes

  Follow this format:
  {{
    "knowledge_type": "{knowledge_type}",
    "nodes": [...],
    "edges": [...]
  }}}}

### Schema

- Node Types
  - User
  - Knowledge
  - Pattern
  - Object
  - Location

- Edge Types
  - Hierarchical Edge
    - User → Knowledge (relation: owns)
    - Knowledge → Pattern (relation: entails)
    - Pattern → Object (relation: target)
    - Pattern → Location (relation: target)
    - Knowledge → Object (relation: alias_of / target_object)
    - Knowledge → Location (relation: target_location)
  - Temporal Edge
    - $Pattern_n$ → $Pattern_{n+1}$ (relation: before)
### Examples

[Example 1]
Instruction: {instruction_example}
Knowledge Type: {knowledge_type_example}
Output: {graph_example}
...

Instruction: {instruction}
Knowledge Type: {knowledge_type}
Output:

