# OpenReview forum: "Embodied Agents Meet Personalization: Investigating Challenges and Solutions Through the Lens of Memory Utilization"
_ICLR.cc/2026/Conference — ICLR 2026 Poster_

### Official Review · Reviewer_t5nh · 2025-10-28

**Soundness:** 3
**Presentation:** 3
**Contribution:** 3
**Rating:** 6
**Confidence:** 4

**Summary:**

The paper provides an empirical analysis of memory utilization in embodied agents by evaluating them on the task of room rearrangement. It involves a memory acquisition stage and then a memory utilization stage. The model is evaluated along two dimensions of memory utilization: the ability to recall object semantics, and the ability to recall a user’s behavioral patterns. The work presents various controlled analyses to isolate the effect of memory on downstream performance, and highlights that current LLM-powered embodied agents can be good at retrieving information about object semantics, but struggle with aggregating multi-episodic memory to understand behavior patterns. Finally, they propose a graph-like structure to summarize user pattern behavior, and show that it improves performance across single-memory and multi-memory tasks, highlighting the need for better memory structure to enable personalized embodied agents.

**Strengths:**

- The work addresses an interesting problem of personalization of embodied agents. Eventually, household agents would more likely than not exist around a user, and would require specializing to their needs. Memory is an interesting and useful component; the analysis done in this work can be useful guidance for future work.
- The paper is very well written and makes it easy to follow and understand the results and the analysis presented.
- I liked the idea of the graph-based hierarchical memory design to embed personal user behaviors; it seems to show clear improvement, at least in the current setup.

**Weaknesses:**

- In Section 5.1, an analysis is conducted, which ablates the value of k, which is the number of retrieved memories. It seems to show that as the value increases how the performance of various agents decreases, credit to information overload.
  - However, consider this: the overload is also indicative of the LLM’s lack of a longer context, which, once it goes beyond a certain number of memories, is unable to grasp and summarize this information, and hence resorts to general semantic knowledge, instead of personalized one.
  - If the above were possible, perhaps a better way of measuring information overload could be to provide a more concise version of relevant information, and only increase the amount of direct clues about the user’s personal behavior, without increasing the information in-context too much.

**Questions:**

- What happens if gold memory is not included? Can it work from noisy memory, could be useful analysis to know.

---

> ### Author Response · Authors · 2025-11-18
> **Response to Reviewer t5nh (Part 1)**
>
> First of all, thank you for recognizing the contributions of our work! We appreciate your thoughtful feedback and would like to address you concerns and questions below:
>
> ## W1
>
> ### W1.1
> > the overload is also indicative of the LLM’s lack of a longer context, which, once it goes beyond a certain number of memories, is unable to grasp and summarize this information, and hence resorts to general semantic knowledge, instead of personalized one.
>
> Thank you for the insightful feedback! We understand your concern as follows: LLM-powered embodied agents may suffer from information overload because LLMs might lack ability to correctly reference information in long contexts.
>
> Therefore, we conduct an experiment in a QA format to evaluate whether LLMs can properly reference information from long contexts.
>
> **Experiment setup.**
> We provided LLMs with the episodic memory information required for personalized assistance tasks along with the corresponding task instructions. We then prompted the models: "Identify the personalized knowledge necessary to perform this instruction based on the context provided below." To evaluate the responses, we used GPT-4o as an LLM-as-a-Judge, supplying it with the stored gold memory and task instruction as references to assess whether the LLM's identified personalized knowledge aligns with that in the gold memory.
>
> | Model           | k=1    | 3       | 5       | 7       | 10      | 20      |
> |-----------------|--------|---------|---------|---------|---------|---------|
> | GPT-4o          | 100%   | 98.71%  | 98.28%  | 96.98%  | 96.98%  | 95.69%  |
> | Qwen-2.5-72b    | 100%   | 93.97%  | 94.40%  | 93.53%  | 93.10%  | 83.19%  |
>
> **Results.**
> As shown in the Table, all models demonstrate high accuracy in correctly identifying personalized knowledge in the QA format. Compared to the results in Section 5.1 Figure 6, this indicates that current LLMs have the ability to reference information from long contexts, but encounter difficulties when applying this information to planning tasks.
>
> We hope this experimental result mitigates your concern. We have included this analysis in Appendix D.5 of the revised manuscript.
>
> ### W1.2
> > If the above were possible, perhaps a better way of measuring information overload could be to provide a more concise version of relevant information, and only increase the amount of direct clues about the user’s personal behavior, without increasing the information in-context too much.
>
>
> Thank you for the recommendation on how to analyze information overload. Through the experiments conducted based on your earlier feedback, we confirmed that LLMs achieve high performance in long-context settings when the task objective is simply to identify personalized knowledge. We believe that this result  further strengthens the justification for our current experimental setup in Section 5.1 for measuring information overload.
>
> Additionally, through Section 6.1 'Investigating Memory Format Simplification', we demonstrate that full trajectories are essential modules for LLM-powered embodied agents due to their in-context learning effects. While the purpose differs, your suggestion that providing only relevant information and direct clues aligns with our motivation for proposing user profile memory to improve performance in Section 6.2.
>
> Please feel free to share if you have any additional questions or concerns regarding this point!

---

> ### Author Response · Authors · 2025-11-18
> **Response to Reviewer t5nh (Part 2)**
>
> ## Q1
> > What happens if gold memory is not included? Can it work from noisy memory, could be useful analysis to know.
>
> Thank you for the valuable question about the effect of different memory conditions. In our baseline setting, the default retrieval setup achieved 96% recall. Under this condition, GPT-4o exhibited a modest performance drop, with Percent Complete showing a decrease of approximately 3.2pp and Success Rate  decrease of approximately 3.0pp.
>
> To further understand how different memory conditions influence performance, we conducted two additional analyses:
> Measuring robustness when noisy or corrupted memory is provided.
> Analyzing how memory quality affects task performance.
>
> ---
>
> ### Measuring robustness when noisy or corrupted memory is provided
>
> First, we conduct an evaluation under noisy and incomplete memory conditions. As the target model, we leverage GPT-4o, and we organize the experimental setup into two representative degradation types: Noisy Memory and Partial Memory. For Noisy Memory, we consider two conditions: (i) random step injection, where unrelated steps from other scenes (30%) are inserted into the memory trajectory to simulate retrieval or perception noise, and (ii) temporal shuffle, where the step order within an episode is randomly permuted to disrupt temporal coherence. For Partial Memory, we examine two forms of incompleteness: (i) random step removal, where 20% of steps are randomly removed to test agent robustness under incomplete recall, and (ii) action-type-specific removal, where all steps of a given action type (Pick, Place, Navigate) are dropped to analyze their impact on planning.
>
> | Model   | Memory  | Memory Type | Planning Cycles ↓ | Sim Steps ↓ | Percent Complete ↑ | ΔPC  | Success Rate ↑ | ΔSR  |
> |---------|---------|-------------|--------------------|-------------|----------------------|------|------------------|------|
> | **GPT-4o** | Baseline | -         | 14.5               | 2481.2      | 85.29%               | -    | 80.60%          | -    |
> |          | Noisy   | Random    | 14.6               | 2759.0      | 81.45%               | -3.8 | 78.61%          | -2.0 |
> |          |         | Shuffle    | 14.4               | 2450.9      | 84.01%               | -1.3 | 80.60%          | 0.0  |
> |          | Partial | Random     | 14.5               | 2526.9      | 84.38%               | -0.9 | 80.10%          | -0.5 |
> |          |         | w/o Pick   | 14.6               | 2555.7      | 82.92%               | -2.4 | 79.10%          | -1.5 |
> |          |         | w/o Place  | 14.5               | 2569.8      | 84.43%               | -0.9 | 82.59%          | +2.0 |
> |          |         | w/o Nav    | 15.4               | 2526.8      | 81.38%               | -3.9 | 78.50%          | -2.1 |
>
> As shown in the Table, the agent remains robust under most forms of incomplete memory, exhibiting only minor degradation. However, two conditions consistently lead to substantial performance drops: (i) the injection of random, noisy entries and (ii) the absence of navigation-related information. This suggests that the agent is more vulnerable to misleading or corrupted inputs than to simple omissions, and that navigation cues play a disproportionately important role in effective planning.

---

> ### Author Response · Authors · 2025-11-18
> **Response to Reviewer t5nh (Part 3)**
>
> ### Analyzing how memory quality affects task performance
>
> Second, to examine how memory quality affects task performance, we run multiple acquisition stages with a GPT-4o based embodied agent and construct a high-quality memory by collecting only the shortest successful trajectories. We then evaluate how much performance improves, relative to the baseline, when different agents perform tasks using this high-quality memory.
>
> | Model | Memory | Stage | Planning Cycles ↓ | Sim Steps ↓ | Percent Complete ↑ | Success Rate ↑ |
> | --- | --- | --- | --- | --- | --- | --- |
> | **GPT-4o** | high-quality | Single Memory | 14.5 | 2481.2 | 85.3% | 80.6% |
> |  |  | Joint Memory | 25.8 | 3619.4 | 83.3% | 63.9% |
> |  | baseline | Single Memory | 16.1 | 2450.8 | 87.9% | 85.1% |
> |  |  | Joint Memory | 28.9 | 3480.7 | 86.7% | 63.9% |
> | **Claude-3.5-Sonnet** | high-quality | Single Memory | 14.1 | 2266.4 | 77.2% | 70.1% |
> |  |  | Joint Memory | 24.2 | 3226.6 | 69.9% | 38.9% |
> |  | baseline | Single Memory | 15.3 | 2258.8 | 69.3% | 63.7% |
> |  |  | Joint Memory | 27.8 | 3198.8 | 64.6% | 33.3% |
> | **Qwen-2.5-72b** | high-quality | Single Memory | 15.2 | 2630.4 | 74.7% | 68.2% |
> |  |  | Joint Memory | 26.4 | 4082.1 | 64.8% | 38.9% |
> |  | baseline | Single Memory | 17.5 | 2691.2 | 72.6% | 67.2% |
> |  |  | Joint Memory | 31.3 | 4027.1 | 68.9% | 36.1% |
> | **Llama-70b** | high-quality | Single Memory | 15.1 | 2364.0 | 72.6% | 67.2% |
> |  |  | Joint Memory | 27.1 | 3374.5 | 64.1% | 27.8% |
> |  | baseline | Single Memory | 19.0 | 2566.6 | 72.2% | 66.7% |
> |  |  | Joint Memory | 31.4 | 3425.2 | 51.3% | 8.3% |
> | **Llama-8b** | high-quality | Single Memory | 16.9 | 2739.0 | 65.1% | 59.2% |
> |  |  | Joint Memory | 26.4 | 3753.2 | 54.0% | 13.9% |
> |  | baseline | Single Memory | 19.0 | 3131.7 | 48.1% | 35.0% |
> |  |  | Joint Memory | 27.4 | 3478.2 | 35.3% | 8.3% |
> | **Qwen-2.5-7b** | high-quality | Single Memory | 15.7 | 2991.5 | 55.7% | 46.3% |
> |  |  | Joint Memory | 23.9 | 4402.0 | 34.0% | 8.3% |
> |  | baseline | Single Memory | 21.8 | 3271.4 | 39.1% | 27.4% |
> |  |  | Joint Memory | 26.9 | 4148.6 | 33.7% | 5.6% |
>
> The results show that in single memory tasks, large models perform similarly to their self-acquired memory baseline, whereas smaller models exhibit substantial gains when provided with high-quality memory.  A similar pattern emerges in joint-memory tasks: while some large models show modest improvements, smaller models benefit much more noticeably, indicating that the influence of optimal trajectories becomes stronger when multiple memories must be integrated. High-quality memory also consistently reduces planning-related steps, reflecting improved planning efficiency. Taken together, these findings emphasize the importance of obtaining optimal trajectories for future embodied-agent systems.Both experimental results can be found in Appendix E.2 of the revised manuscript.

---

> > ### Author Response · Authors · 2025-11-26
> >
> > Dear Reviewer t5nh,
> >
> > Thank you again for your time and effort to provide your insightful feedback on our paper.
> >
> > We have addressed your comments in the responses above. If you have any remaining questions or require further clarification, we would be happy to address them before the time window closes.
> >
> > Thank you so much for your time and valuable feedback!
> >
> > Best regards,
> >
> > The Authors of Paper 16591

---

### Official Review · Reviewer_f2sq · 2025-11-01

**Soundness:** 3
**Presentation:** 3
**Contribution:** 2
**Rating:** 4
**Confidence:** 3

**Summary:**

The paper investigates how LLM-powered embodied agents handle personalization via memory, focusing on two dimensions: recognizing objects with user-specific semantics and recalling sequences of user behavioral patterns. The authors propose an evaluation framework named MEMENTO, including both single-memory and joint-memory tasks, and find that while agents can use simple object semantic memories, they struggle to incorporate sequential memory of user routines into planning. A deeper analysis identifies two main bottlenecks: information overload when too many memories are retrieved, and coordination failures when multiple memories must be integrated. To address these, the paper explores memory architectures and proposes a hierarchical knowledge-graph-based user-profile memory module that manages personalized knowledge separately.

**Strengths:**

- The paper provides a systematic analysis of the memory utilization of LLM-powered agents for personalized scenarios, including multiple personal preferences.
- Underperformance even with the top-k most relevant information for a specific personalization seems interesting.
- The quantitative comparisons are conducted through diverse open- and closed-sourced LLMs.

**Weaknesses:**

- It is unclear why the problem addressed in this paper is specifically related to embodied agents. Given that the issues (Sec. 3) and underperformances (Sec. 4 and 5) come from the incapabilities of LLMs, it looks more related to LLMs than embodied agents.
- As it is explicitly given how some objects and routines are to be described, the problem addressed in this paper is rather related to using the context, which should not necessarily have to be personalization, provided well. In this sense, the term "personalization" seems misleading.
- For the result in Figure 4, the authors argue that the current embodied agents struggle to comprehend user patterns, but I'm not fully convinced because the performance drops may come from the longer horizon of the user patterns than the one of the object semantics. This needs to be distinguished. In addition, as there is no clear performance gaps in object semantics, it is unclear if LLMs indeed suffers from using personalized information.
  - The result in Figure 3 is somewhat related to this point for original/acquisition to single/joint memory utilization.
- The evaluation metrics used in this paper seem to be inappropriate. The problem that the authors tackle in this paper is about how well LLMs can use personalized information given in the context, but the metrics used are about how well they complete the entire tasks, potentially including aspects beyond personalized memory utilization.

**Questions:**

- For the joint memory, how many personalizations can be jointly used for some decent performance? Is there any quantitative analysis of the number of required personalizations with the corresponding performance of the downstream tasks?
- Can the analysis be extended to scenarios where personalization indicators (e.g., "That's my favoriate ..." or "It's my ...") are not given? Addressing personalized information obtained from a set of videos might be a more practical scenario.
- In Figure 6, some models improve when taking more retrieved samples. Why is this the case? As the current personalization addresses the mapping from an expression to a target object or a set of user patterns, where a single sample might be sufficient for their description.

---

> ### Author Response · Authors · 2025-11-18
> **Response to Reviewer f2sq (part1)**
>
> We truly appreciate the time you spent reviewing our work and providing feedback! It appears there might be some differences in perspective or misunderstandings on a few points. We will provide our perspective and additional explanations regarding the issues you raised, and would be very grateful if you feel comfortable sharing any additional points you would like to discuss so we can have a constructive discussion!
>
> ## W1
> > It is unclear why the problem addressed in this paper is specifically related to embodied agents. Given that the issues (Sec. 3) and underperformances (Sec. 4 and 5) come from the incapabilities of LLMs, it looks more related to LLMs than embodied agents.
>
> We understand your concern that since most limitations discussed in the paper relates to LLMs, the connection to embodied agents may not be immediately apparent. However, we believe that the knowledge and capabilities required from LLMs differ depending on the task they address and the role they assume. As discussed in the first paragraph of the Introduction and the "LLM-powered embodied agents" paragraph in the Related Work section, the field has increasingly leveraged LLMs' natural language understanding and planning capabilities to solve embodied tasks. Therefore, the limitations exhibited by LLM-powered embodied agents when performing embodied tasks—including knowledge about the physical world, planning abilities, and the capacity to utilize memory for personalized assistance—are inseparable from the embodied agent context.
>
> We hope our perspective has been clearly conveyed, and please feel free to share if you would like to discuss this point further.
>
>
> ## W2
> > As it is explicitly given how some objects and routines are to be described, the problem addressed in this paper is rather related to using the context, which should not necessarily have to be personalization, provided well. In this sense, the term "personalization" seems misleading.
>
> Thank you for raising this point about the term "personalization." However, we believe there is an important distinction to clarify: while context utilization doesn't have to be personalization, personalization requires context utilization. As stated in the second paragraph of the Introduction, our paper specifically targets personalized assistance in LLM-powered embodied agents. To provide personalized assistance in embodied agents, memory utilization capability is required, and memory utilization in LLMs fundamentally involves context utilization. Therefore, since our paper primarily focuses on personalized assistance, we believe the term "personalization" accurately reflects our main objective and is not misleading.
>
> We hope this addresses your concern, and are happy to elaborate further if needed.
>
> ## W3
>
> ### W3.1
> > For the result in Figure 4, the authors argue that the current embodied agents struggle to comprehend user patterns, but I'm not fully convinced because the performance drops may come from the longer horizon of the user patterns than the one of the object semantics. This needs to be distinguished.
>
>
> We appreciate your feedback on raising this point about the underlying cause of the performance drop. To clarify whether the higher difficulty of user pattern tasks arises solely from their longer horizons, we conducted an analysis of the underlying structure of both personalized knowledge types.
> We compared the number of (object, location) pairs that must be satisfied for successful completion of both tasks.
>
> | Number of (Object, Location) Pairs | Object Semantics | User Pattern |
> | :---: | :---: | :---: |
> | 1 | 4.5% | 6.3% |
> | 2 | 65.2% | 57.1% |
> | 3 | 24.7% | 28.6% |
> | 4 | 4.5% | 5.4% |
> | 5 | 0.0% | 0.9% |
> | 6 | 1.1% | 1.8% |
>
> The Table reports the percentage of instructions requiring each number of (object, location) pairs for both task types. As shown, the percentages across corresponding counts are nearly identical. This indicates that the performance drop observed in our experiments is more likely due to the intrinsic characteristics of the task itself rather than the longer planning horizon that the task requires. We have included this analysis in Appendix C.3.4.

---

> ### Author Response · Authors · 2025-11-18
> **Response to Reviewer f2sq (part2)**
>
> ### W3.2
> > In addition, as there is no clear performance gaps in object semantics, it is unclear if LLMs indeed suffers from using personalized information. The result in Figure 3 is somewhat related to this point for original/acquisition to single/joint memory utilization.
>
> Thank you for raising this important point. We would like to clarify that our argument about LLMs struggling with personalized information is based on the performance drop observed in Table 1 when comparing the memory utilization stage to the memory acquisition stage. As stated in the first paragraph of Section 4.2 at line 300-304, this performance degradation demonstrates that LLMs have difficulty effectively utilizing personalized knowledge from an overall task perspective.
>
>
> Furthermore, through Figure 4, we explicitly demonstrated in the second paragraph of Section 4.2—as well as in the Introduction—that among the two types of personalized knowledge comprising a single task in the memory utilization stage, LLMs are effective with object semantics but suffer with user patterns. We provided this analysis to convey why LLMs struggle when utilizing personalization information. Therefore, we believe the performance gap is clear, particularly evident in the user pattern component.
>
> We hope this clarifies our position, and welcome any further discussion on this matter.
>
> ## W4
> > The evaluation metrics used in this paper seem to be inappropriate. The problem that the authors tackle in this paper is about how well LLMs can use personalized information given in the context, but the metrics used are about how well they complete the entire tasks, potentially including aspects beyond personalized memory utilization.
>
> We appreciate your feedback on our evaluation metrics. We agree that evaluating memory utilization for personalized assistance requires a more targeted approach than simply measuring entire task completion. This is precisely why, as mentioned in the third paragraph of the Introduction and Section 3.3 line 201-202, we proposed Memento as our evaluation framework to quantify memory utilization capability. Since we aim to assess memory utilization from the perspective of whether LLM-powered embodied agents can effectively provide personalized assistance, we believe showing entire task evaluation metrics is necessary. Furthermore, through Memento, we quantify memory utilization ability in a way that minimizes—though does not perfectly isolate—the influence of other aspects. We consider this to be the main contribution of our evaluation framework and hope this addresses your concern.
>
> ## Q1
> > For the joint memory, how many personalizations can be jointly used for some decent performance? Is there any quantitative analysis of the number of required personalizations with the corresponding performance of the downstream tasks?
> Thank you for pointing out the scalability of personalization. To answer this, we perform a fine-grained analysis that quantifies how specific personalization components affect downstream task performance across four models.
> At first, we focused on object semantics, which require the agent to recall the specific object information from memory. We examine how performance varies with the number of target objects, analyzing whether increasing object count affects overall performance.
>
> | Model | Object | PC | SR | Task Number |
> | :---: | :---: | :---: | :---: | :---: |
> | GPT-4o | 1 | 92.9% | 92.9% | 14 |
> |  | 2 | 93.9% | 90.4% | 52 |
> |  | 3 | 97.5% | 91.3% | 23 |
> | Qwen-2.5-72b | 1 | 83.3% | 78.6% | 14 |
> |  | 2 | 87.6% | 80.8% | 52 |
> |  | 3 | 89.0% | 82.6% | 23 |
> | Llama-3.1-8b | 1 | 67.9% | 64.3% | 14 |
> |  | 2 | 66.0% | 51.9% | 52 |
> |  | 3 | 56.8% | 26.1% | 23 |
> | Qwen-2.5-7b | 1 | 64.3% | 57.1% | 14 |
> |  | 2 | 45.4% | 28.9% | 52 |
> |  | 3 | 33.0% | 8.7% | 23 |
>
> As shown in the above Table, larger models remained robust regardless of the number of target objects, whereas smaller models showed a sharp performance drop as the object count increased. This indicates that increasing object count primarily challenges lower-capacity models, whereas higher-capacity models can reliably handle richer object information.
>
> Next, for the user pattern, which requires retrieving the whole (object, location) pairs via memory. To understand what makes these tasks particularly challenging, we examined how the number of objects and locations in each instruction influences model performance.

---

> > ### Author Response · Authors · 2025-11-18
> > **Response to Reviewer f2sq (part3)**
> >
> > | Model | Object | Location | PC | SR | Task Number |
> > | :---: | :---: | :---: | :---: | :---: | :---: |
> > | GPT-4o | 1 | 1 | 85.7% | 85.7% | 7 |
> > |  | 1 | 2 | 64.3% | 57.1% | 7 |
> > |  | 2 | 2 | 85.4% | 84.2% | 57 |
> > |  | 3 | 3 | 94.8% | 90.6% | 32 |
> > |  | 2 | 4 | 33.3% | 33.3% | 6 |
> > |  | 3 | 5 | 0.0% | 0.0% | 1 |
> > |  | 3 | 6 | 50.0% | 50.0% | 2 |
> > | Qwen-2.5-72b | 1 | 1 | 57.1% | 57.1% | 7 |
> > |  | 1 | 2 | 35.7% | 28.6% | 7 |
> > |  | 2 | 2 | 62.6% | 59.7% | 57 |
> > |  | 3 | 3 | 74.1% | 65.6% | 32 |
> > |  | 2 | 4 | 33.3% | 33.3% | 6 |
> > |  | 3 | 5 | 0.0% | 0.0% | 1 |
> > |  | 3 | 6 | 18.8% | 0.0% | 2 |
> > | Llama-3.1-8b | 1 | 1 | 35.7% | 28.6% | 7 |
> > |  | 1 | 2 | 21.4% | 0.0% | 7 |
> > |  | 2 | 2 | 38.0% | 35.1% | 57 |
> > |  | 3 | 3 | 43.3% | 21.9% | 32 |
> > |  | 2 | 4 | 10.0% | 0.0% | 6 |
> > |  | 3 | 5 | 0.0% | 0.0% | 1 |
> > |  | 3 | 6 | 18.8% | 0.0% | 2 |
> > | Qwen-2.5-7b | 1 | 1 | 28.6% | 28.6% | 7 |
> > |  | 1 | 2 | 28.6% | 28.6% | 7 |
> > |  | 2 | 2 | 37.4% | 28.1% | 57 |
> > |  | 3 | 3 | 40.0% | 31.3% | 32 |
> > |  | 2 | 4 | 0.0% | 0.0% | 6 |
> > |  | 3 | 5 | 0.0% | 0.0% | 1 |
> > |  | 3 | 6 | 18.8% | 0.0% | 2 |
> >
> > Across all four models, we observed a consistent trend that tasks where a single object maps to multiple locations exhibit substantial performance drops. This setting requires the agent to retrieve and apply multiple location bindings for the same object across separate steps, increasing the sequential recall burden. Conversely, when the number of objects and locations is balanced (e.g., 1 object - 1 location or 2 objects - 2 locations), performance remains relatively high. This suggests that the main source of difficulty is the sequential nature of recalling multiple location bindings for the same object, which strains the model’s memory utilization capabilities.
> >
> > ## Q2
> > > Can the analysis be extended to scenarios where personalization indicators (e.g., "That's my favoriate ..." or "It's my ...") are not given? Addressing personalized information obtained from a set of videos might be a more practical scenario.
> >
> > We appreciate the insightful question. We agree that real-world human-agent communication often involves indirect or ambiguous references. Therefore, we conduct experiments to assess whether current LLM-powered embodied agents can interpret ambiguous instructions that indirectly refer to previously encountered personalized knowledge.
> >
> > **Experiment setup.**
> > We create 30 episodes by random sampling and manually modify these episodes to include indirect references to personalized knowledge through contextual cues, synonyms, or causal references (e.g., Can you set my afternoon teatime routine? \-\> I’m about to enjoy my afternoon tea. Could you set things up as I like?).
> >
> > | Model | Percent Complete (%) | Success Rate (%) |
> > | :---- | :---- | :---- |
> > | GPT-4o  | 92.0 | 90.0 |
> > | GPT-4o (Indirect instructions) | 80.4 | 73.3 |
> > | Qwen-2.5-72b  | 75.1 | 66.7 |
> > | Qwen-2.5-72b (Indirect instructions) | 59.6 | 53.3 |
> >
> > **Result.**
> > As shown in the Table, LLM-powered embodied agents demonstrate additional performance degradation compared to original (direct) instructions when facing ambiguous instructions. This result indicates that improvements are required, whether through enhanced LLM capabilities or systematic approaches, to better understand user intention. We have included these results in Appendix E.4.2.
> >
> > ## Q3
> > > In Figure 6, some models improve when taking more retrieved samples. Why is this the case? As the current personalization addresses the mapping from an expression to a target object or a set of user patterns, where a single sample might be sufficient for their description.
> >
> > Thank you for pointing out this interesting phenomenon in our experimental results. We attribute the performance improvement from k=1 to k=5 in object semantics to the in-context learning effect. As demonstrated in the findings of Section 6.1, full trajectories in episodic memory provide in-context learning benefits for LLMs, leading to performance gains. For user patterns, we observe performance degradation, which we believe occurs because the failure to sufficiently reference personalized knowledge outweighs the performance gains from in-context learning.
> >
> > Furthermore, while we agree that a single gold memory sample might be sufficient for providing personalized assistance, we believe additional samples are necessary from two perspectives: (1) As shown in Section 6.1, episodic memory serves not only to provide personalized knowledge but also as memory for in-context learning. (2) Although our experiments were conducted in a setting where gold memory is always included to evaluate memory utilization capability, as mentioned in Section 3.4, recall is 96.5% at k=5 and 86.1% at k=3. This indicates that accurate memory cannot always be referenced, suggesting that the number of retrieved memories must be carefully considered in real-world setups.

---

> ### Author Response · Authors · 2025-11-26
>
> Dear Reviewer f2sq,
>
> Thank you again for your time and effort to provide your insightful feedback on our paper.
>
> We have addressed your comments in the responses above. If you have any remaining questions or require further clarification, we would be happy to address them before the time window closes.
>
> Thank you so much for your time and valuable feedback!
>
> Best regards,
>
> The Authors of Paper 16591

---

### Official Review · Reviewer_KG3h · 2025-11-01

**Soundness:** 3
**Presentation:** 4
**Contribution:** 3
**Rating:** 6
**Confidence:** 3

**Summary:**

This paper investigates the challenges LLM-powered embodied agents face in performing personalized assistance tasks, which require leveraging user-specific knowledge from past interactions (i.e., episodic memory). The authors introduce MEMENTO, a novel two-stage evaluation framework designed to quantify an agent's memory utilization capabilities. This framework assesses performance along two key dimensions: 'object semantics' (recalling personal meaning of objects) and 'user patterns' (recalling behavioral routines). Through experiments with a suite of modern LLMs (e.g., GPT-4o, Llama-3.1) , the authors find that while agents can recall simple object semantics, they significantly struggle to apply sequential user patterns. They identify two critical bottlenecks: 'information overload' from large retrieved memory sets and 'coordination failures' when handling multiple memories. Based on these findings, the paper proposes a hierarchical knowledge graph-based user-profile memory module that separates structured personalized knowledge from raw episodic memory, demonstrating substantial performance improvements.

**Strengths:**

- The two-stage design (memory acquisition vs. utilization) provides a principled way to isolate memory effects from other capabilities, with identical goals but varying instructions across stages, enabling precise measurement of memory utilization through metrics.

- The dataset construction process is rigorous with multiple quality controls.

- The experimental analysis is comprehensive with insightful qualitative analysis

**Weaknesses:**

- The paper introduces the personalized object rearrangement task as a Partially Observable Markov Decision Process in Section 3.1. The actual agent implementation, however, is a hierarchical controller that uses an LLM as a high-level policy planner in a ReAct-style prompting format. The POMDP formalism and its associated equations feel disconnected from the practical, prompt-based system that is actually built and evaluated. The formalism is not explicitly used to derive the agent architecture or learning algorithm (as it's a zero-shot planner). This creates a minor mismatch between the stated mathematical grounding and the implementation.

- The simplification of the full embodied agent problem means the episodic memory ($h_{acq}$) stored is "clean." It's unclear how performance would be affected if the memory traces themselves were noisy due to perception errors. The proposed KG solution, which relies on clean, symbolic entity names, would also be challenging to implement without this oracle.

- Missing analysis of how memory grows over time (all experiments use fixed 1-2 reference episodes)

**Questions:**

- What if object semantics and user patterns were stored as structured text templates rather than knowledge graphs? Would this achieve similar benefits with lower complexity?

---

> ### Author Response · Authors · 2025-11-18
> **Response to Reviewer KG3h (part1)**
>
> We appreciate your thoughtful review and recognition of our efforts on designing and constructing our evaluation framework\! We hope that the following responses help clarify the reviewer’s questions:
>
> ## W1
> > …  The formalism is not explicitly used to derive the agent architecture or learning algorithm (as it's a zero-shot planner). This creates a minor mismatch between the stated mathematical grounding and the implementation.
>
> We sincerely apologize for any confusion regarding the POMDP formulation. As you pointed out, the POMDP formulation  is not used to derive the agent architecture or learning algorithm, but to facilitate understanding of the personalized assistance task environment. Unlike MDP formulation which assumes fully observable states, we adopted POMDP to accurately represent the partial observability of our task environment. In our task setup, the agent must navigate to specific spaces to discover object locations and acquire information, and visual clues can only be obtained through actions selected by the LLM policy via the perception skill library. The POMDP formulation was intended to capture these key characteristics of our task environment.
>
> We hope this addresses your concern. We have revised lines 139-140 in Section 3.1 to provide clarification on partial observability. Please feel free to share any additional feedback you may have.
>
> ## W2
>
> ### W2.1
> > The simplification of the full embodied agent problem means the episodic memory () stored is "clean." It's unclear how performance would be affected if the memory traces themselves were noisy due to perception errors.
>
> Thank you for pointing out this concern about how performance might degrade when episodic memory contains noise or partial information, which differs from the clean setup used in MEMENTO. To provide insights into whether current LLM-powered embodied agents are robust to noisy memory, we conducted an analysis under noisy and incomplete memory conditions.
>
> As the target model, we leverage GPT-4o, and we organize the experimental setup into two representative degradation types: *Noisy Memory* and *Partial Memory*. For *Noisy Memory*, we consider two conditions: (i) random step injection, where unrelated steps from other scenes (30%) are inserted into the memory trajectory to simulate retrieval or perception noise, and (ii) temporal shuffle, where the step order within an episode is randomly permuted to disrupt temporal coherence. For *Partial Memory*, we examine two forms of incompleteness: (i) random step removal, where 20% of steps are randomly removed to test agent robustness under incomplete recall, and (ii) action-type-specific removal, where all steps of a given action type (Pick, Place, Navigate) are dropped to analyze their impact on planning.
>
> | Model | Memory | Memory Type | Planning Cycles ↓ | Sim Steps ↓ | Percent Complete ↑ | ∆PC | Success Rate ↑ | ∆SR |
> | :---: | ----- | :---: | :---: | :---: | :---: | :---: | :---: | :---: |
> |  | Gold | Baseline | 14.5 | 2481.2 | 85.29% | \- | 80.60% | \- |
> | GPT-4o | Noisy  | Random | 14.6 | 2759.0 | 81.45% | **\-3.8** | 78.61% | **\-2.0** |
> |  |  | Shuffle | 14.4 | 2450.9 | 84.01% | \-1.3 | 80.60% | 0.0 |
> |  | Partial  | Random | 14.5 | 2526.9 | 84.38% | \-0.9 | 80.10% | \-0.5 |
> |  |  | w/o Pick | 14.6 | 2555.7 | 82.92% | \-2.4 | 79.10% | \-1.5 |
> |  |  | w/o Place | 14.5 | 2569.8 | 84.43% | \-0.9 | 82.59% | \+2.0 |
> |  |  | w/o Nav | 15.4 | 2526.8 | 81.38% | **\-3.9** | 78.50% | **\-2.1** |
>
>
> As shown in the Table, the agent remains robust under most forms of incomplete memory, exhibiting only minor degradation. However, two conditions consistently lead to substantial performance drops: (i) the injection of random, noisy entries and (ii) the absence of navigation-related information. This suggests that the agent is more vulnerable to misleading or corrupted inputs than to simple omissions, and that navigation cues play a disproportionately important role in effective planning. We have included these results in Appendix E.2.2.
>
>
> ### W2.2
>
> > The proposed KG solution, which relies on clean, symbolic entity names, would also be challenging to implement without this oracle.
>
> We also want to thank the reviewer for raising a valid point regarding the potential issue where our proposed User Profile Memory may appear to depend on clean, symbolic entity names.
>
> To directly evaluate the robustness of our proposed User Profile Memory under realistic conditions, we additionally conduct a qualitative analysis on how well User Profile Memory handles natural noise that arises in real user interactions.
>
> We identified two noise types that naturally arise when users interact with a personalized embodied agent:
>
> 1. **Knowledge Update Noise** \- the user expresses an existing knowledge using a different phrase (synonyms, paraphrases, alias names).
> 2. **Node Reference Noise** \- new knowledge refers to an existing object / location using a different form.

---

> > ### Author Response · Authors · 2025-11-18
> > **Response to Reviewer KG3h (part2)**
> >
> > For each category, three undergraduate annotators created 10 noisy instructions grounded in the scene of MEMENTO, ensuring the noise realistically reflects how humans describe object semantics or user patterns.
> >
> > The table below shows representative examples:
> >
> > | Original Knowledge | Modified Knowledge |
> > | :---: | :---: |
> > | Family heirloom vase | Inherited family vase |
> > | Bedtime setup | Nighttime setup before going to bed |
> >
> > | Original Reference | Modified Reference |
> > | :---: | :---: |
> > | Black teapot with curved spout and ornate lid | **A curved-spout black teapot featuring an intricately decorated lid** |
> > | Move the **toy airplane** from the bedroom to **the living room table**. Place it next to the **couch**.  | Relocate the **toy plane** from the bedroom onto **the** **table in the living room,** positioning it next to the **sofa.** |
> >
> > **Results**
> >
> > |  | Accuracy |
> > | ----- | :---: |
> > | Knowledge Update Noise | 100% |
> > | Node Reference Noise | 75% |
> >
> >
> > The results show that knowledge update/add decisions are handled reliably across all samples. These findings indicate that our method handles natural linguistic noise reasonably well. However, node reference shows a few missed cases (5 out of 20).
> >
> > **Failure cases**
> >
> > | Existing Node | Add Node |
> > | :---: | :---: |
> > | mouse pad | **mouse mat** |
> > | clock | **alarm clock** |
> > | android figure which is a black and white android panda figure | **black and white Android Panda Figurine** |
> > | kitchen counter | **countertop at kitchen** |
> > | another shelf | **separate shelf** |
> >
> > Although these missed cases amount to 25%, the model still correctly handles 75% of the reference variations. These failures indicate that reducing unnecessary node duplication remains an important challenge that requires further refinement.
> >
> > We believe this direction is important because mitigating node duplication directly improves memory consistency, which is a key factor influencing downstream task performance in personalized embodied agents. We have included these results in Appendix E.4.4 of the revised manuscript.
> >
> > ## W3
> > > Missing analysis of how memory grows over time (all experiments use fixed 1-2 reference episodes)
> >
> > We apologize for any confusion in our description of the evaluation setup. To clarify, we conduct the memory acquisition stage for all episodes first, then perform top-5 similarity-based retrieval from the accumulated episodic memories. Subsequently, if the episodic memory corresponding to the acquisition stage episode is not included among the top-5 retrieved memories for the memory utilization stage episode, we randomly replace one of the five memories to ensure it is included. Therefore, to answer your question, memory accumulates as episodes are performed during the acquisition stage, and all experiments utilize top-5 retrieval with fixed 1-2 reference episodes.
> >
> > We hope this addresses your concern, and we have revised Section 3.4 'Retrieval Setup' at line 250-251 for better clarification.
> >
> > ## Q1
> > > What if object semantics and user patterns were stored as structured text templates rather than knowledge graphs? Would this achieve similar benefits with lower complexity?
> >
> > We truly appreciate your insightful question! We agree that a structured text template provides a simpler and lightweight alternative for storing semantic knowledge, and we considered this format in the early version of our method.
> >
> > Specifically, we construct the memory using a JSON-based structured text template with three fields (User, Knowledge, Description) and evaluate it across four LLM-based embodied agents.
> >
> > | Model | Setup | PC | SR |
> > | :---: | :---: | :---: | :---: |
> > | GPT-4o | Single (baseline) | 87.9% | 85.1% |
> > |  | Joint (baseline) | 86.7% | 63.9% |
> > |  | **Single (UPM)** | **92.5%** | **90.5%** |
> > |  | **Joint (UPM)** | 87.9% | 75.0% |
> > |  | **Single (text)** | 92.0% | 90.0% |
> > |  | **Joint (text)** | **91.8%** | **80.6%** |
> > | Qwen-2.5-72b | Single (baseline) | 72.6% | 67.2% |
> > |  | Joint (baseline) | 68.9% | 36.1% |
> > |  | **Single (UPM)** | 88.9% | **86.6%** |
> > |  | **Joint (UPM)** | **85.8%** | **72.2%** |
> > |  | **Single (text)** | **89.7%** | **86.6%** |
> > |  | **Joint (text)** | 81.4% | 61.8% |
> > | LLaMA-3.1-8b | Single (baseline) | 48.1% | 35.0% |
> > |  | Joint (baseline) | 35.3% | 8.3% |
> > |  | **Single (UPM)** | **62.8%** | 52.7% |
> > |  | **Joint (UPM)** | 48.3% | 16.7% |
> > |  | **Single (text)** | 61.8% | **53.7%** |
> > |  | **Joint (text)** | **51.3%** | **19.4%** |
> > | Qwen-2.5-7b | Single (baseline) | 39.1% | 27.4% |
> > |  | Joint (baseline) | 33.7% | 5.6% |
> > |  | **Single (UPM)** | 51.9% | 40.8% |
> > |  | **Joint (UPM)** | **40.2%** | **16.7%** |
> > |  | **Single (text)** | **54.4%** | **43.3%** |
> > |  | **Joint (text)** | 33.2% | 11.1% |

---

> > > ### Author Response · Authors · 2025-11-18
> > > **Response to Reviewer KG3h (part3)**
> > >
> > > |  |  | Object semantics |  | User Pattern |  |
> > > | ----- | ----- | :---: | :---: | :---: | :---: |
> > > | Model | Stage | PC | SR | PC | SR |
> > > | GPT-4o | Single (baseline) | 94.7% | 91.0% | 82.6% | 80.4% |
> > > |  | Single (UPM) | **96.6%** | **94.4%** | **89.3%** | **87.5%** |
> > > |  | Single (text) | 96.1% | **94.4%** | 88.7% | 86.6% |
> > > | Qwen-2.5-72b | Single (baseline) | 87.3% | 80.9% | 60.9% | 56.3% |
> > > |  | Single (UPM) | 92.0% | 88.8% | **86.5%** | **84.8%** |
> > > |  | Single (text) | **95.9%** | **93.3%** | 84.7% | 81.3% |
> > > | LLaMA-3.1-8b | Single (baseline) | 63.2% | 46.6% | 36.2% | 25.9% |
> > > |  | Single (UPM) | 60.6% | 47.2% | **64.6%** | **57.1%** |
> > > |  | Single (text) | **63.7%** | **51.7%** | 60.4% | 55.4% |
> > > | Qwen-2.5-7b | Single (baseline) | 45.2% | 28.1% | 34.4% | 26.8% |
> > > |  | Single (UPM) | **45.6%** | **32.6%** | 56.9% | 47.3% |
> > > |  | Single (text) | 42.5% | 28.1% | **63.9%** | **55.4%** |
> > >
> > > As shown in the Tables, the structured text template memory demonstrates competitive performance, showing results comparable to those of the User Personalized Memory. This aligns with your intuition that a lightweight structured format can reduce complexity while still supporting reasonable performance under the diverse llm-based embodied agents.
> > >
> > > However, this approach becomes increasingly limited when considering real-world, long-term deployment. We believe that an embodied agent must continually accumulate and update knowledge through ongoing interaction, maintain consistency across heterogeneous information, and support cross-referencing across components. Therefore, even though a structured text template demonstrates comparable performance with our proposed User Profile Memory, it shows limitations in supporting continual updates, preserving relational consistency, and scaling with long-term personalized interactions.

---

> > > > ### Author Response · Authors · 2025-11-26
> > > >
> > > > Dear Reviewer KG3h,
> > > >
> > > > Thank you again for your time and effort to provide your insightful feedback on our paper.
> > > >
> > > > We have addressed your comments in the responses above. If you have any remaining questions or require further clarification, we would be happy to address them before the time window closes.
> > > >
> > > > Thank you so much for your time and valuable feedback!
> > > >
> > > > Best regards,
> > > >
> > > > The Authors of Paper 16591

---

### Meta-Review · Area_Chair_2X5r · 2026-01-06

**Summary:**

Summary:
This paper studies how LLM-based embodied agents handle personalization that depends on memory from past interactions. The authors introduce MEMENTO, a two-stage evaluation framework that measures memory use along two axes: remembering user-specific object meanings and recalling user behavior patterns over time. Experiments with modern LLMs show that agents can handle simple object semantics but struggle with sequential user routines. The analysis reveals two main problems: too many retrieved memories cause information overload, and agents fail to coordinate multiple memories during planning. To address this, the paper proposes a hierarchical, knowledge-graph–based user profile that separates structured personal knowledge from raw episodic memory, leading to clear performance improvements.

Strengths:
1. The two-stage design provides a principled way to isolate memory effects from other capabilities.
2. Rigorous data constructions with multiple quality controls.
3. Comprehensive experimental analysis.


Weaknesses:
1. The POMDP formulation is not well aligned with the ReAct-style planning implementation.
2. It is unclear how noisy memory may affect the performance.
3. Lack of analysis of how memory grows over time.
4. It is unclear why the problem addressed in the paper is specific to embodied agents.
5. Personalization seems misleading. Context might be a better characterization of the study.
6. The true causes of the performance drop are not entirely clear.
7. Metrics do not evaluate personalization.
8. It would be good to isolate the effect of a longer context on LLM.

**Reviewer Concerns:**

All reviewer concerns have been addressed by the rebuttal.

**Reviewer Scores:**

Reviewer KG3h and t5nh’s scores were already positive. The responses were helpful in addressing the concerns, but it is hard to say whether the reviewers will further raise the score.

Reviewer f2sq may raise their score as I find the responses adequate.

---

### Decision · Program_Chairs · 2026-01-26

Accept (Poster)